# Biogeochemical model Biome-BGCMuSo v6.2 provides plausible and accurate simulations of carbon cycle in Central European beech forests

Katarína Merganičová[1,2], Ján Merganič[3], Laura Dobor[1], Roland Hollós[4,5], Zoltán Barcza[1,4], Dóra Hidy[9], Zuzana Sitková[6], Pavel Pavlenda[6], Hrvoje Marjanovic[8], Daniel Kurjak[3], Michal Bošeľa[3], Doroteja Bitunjac[8], Masa Zorana Ostrogovic Sever[8], Jiří Novák[7], Peter Fleischer[3], Tomáš Hlásny[1]

[1]Czech University of Life Sciences Prague, Faculty of Forestry and Wood Sciences, Kamýcká 129, 16500 Praha 6 – Suchdol, Czech Republic

[2]Department of Biodiversity of Ecosystems and Landscape, Institute of Landscape Ecology, Slovak Academy of Sciences, 949 01 Nitra, Slovak Republic

[3]Technical University in Zvolen, Faculty of Forestry, T. G. Masaryka 24, 960 53 01 Zvolen, Slovak Republic

[4]ELTE Eötvös Loránd University, Department of Meteorology, Pázmány P. sétány 1/A, H-1117 Budapest, Hungary

[5]HUN-REN Centre for Agricultural Research, Agricultural Institute,Brunszvik u. 2, H-2462 Martonvásár, Hungary

[6]National Forest Centre -, Forest Research Institute Zvolen, T. G. Masaryka 2175/22, SK – 960 92 01 Zvolen, Slovak Republic

[7]Forestry and Game Management Research Institute, Research Station at Opočno, Na Olive 550, 517 73 Opočno, Czech Republic

[8]Croatian Forest Research Institute, Division for Forest Management and Forestry Economics, Cvjetno naselje 41, 10450 Jastrebarsko, Croatia

[9]HUN-REN-MATE Agroecology Research Group, 2100 Gödöllő, Páter K. u. 1.

*Correspondence to*: Katarína Merganičová (k.merganicova@forim.sk)

**Abstract.** Process-based ecosystem models are increasingly important for predicting forest dynamics under future environmental conditions, which may encompass non-analogous climate coupled with unprecedented disturbance regimes. However, challenges persist due to the extensive number of model parameters, scarce calibration data, and the trade-offs between the local precision and the applicability of the model over a wide range of environmental conditions. In this paper, we describe a protocol that allows a modeller to collect transferable ecosystem properties based on ecosystem characteristic criteria and to compile the parameters that need to be described in the field.

We applied the procedure to develop a new parameterization for the European beech (*Fagus sylvatica* L.) for the Biome-BGCMuSo model, the most advanced member of the Biome-BGC family. For model calibration and testing, we utilized multiyear forest carbon data from 87 plots distributed across five European countries. The initial values of 48 new ecophysiological parameters were defined based on the literature review. The final values of 6 calibrated parameters were optimised for single sites as well as for multiple sites using the Generalised Likelihood Uncertainty Estimation and model output conditioning that ensured plausible simulations based on user-defined ranges of carbon stock output variables (carbon stock in aboveground wood biomass, soil, and litter) and finding the intersections of site-specific plausible parameter hyperspaces. To support the model use, we tested the model performance in simulating tree aboveground wood, soil, and litter carbon across a large geographical gradient of Central Europe and evaluated the trade-offs between parameters tailored to single plots and parameters estimated using multiple sites.

Our findings indicated that parameter sets derived from single sites provided an improved local accuracy of simulations of aboveground wood, soil, and litter carbons stocks by 35 %, 55 %, and 11 % in comparison to the a priori parameter set. However, their broader applicability was very limited. A multi-site optimised parameter set, on the other hand, performed satisfactorily across the entire geographical domain studied here, including sites not involved in the parameter estimation, but the errors were on average by 26 %, 35 % and 9 % greater for the aboveground wood, soil, and litter carbons stocks than those obtained with the site-specific parameter sets. Importantly, model simulations demonstrated plausible responses across large-scale environmental gradients, featuring a clear production optimum of beech that aligns with empirical studies. These findings suggest that the model is capable of accurately simulating the dynamics of the European beech across its range and can be used for more comprehensive experimentations.

# 1    Introduction

Complex process-based vegetation dynamics models (PBMs) typically contain many parameters specifying physiology, biochemistry, phenology, and allocation patterns of different vegetation types or species (Cameron et al., 2013; van Oijen, 2017). Parameter values are estimated based on different field or laboratory measurements, trial-and-error parameter adjustments or probabilistic methods (Forrester et al., 2021). A comprehensive review of calibration methods can be found in Hollós et al. (2022). Thereby, each measurable parameter has its own variability that emerges from environmental conditions, sampling and measurement errors. Such a value range can be interpreted as a parameter probability distribution or parameter uncertainty (van Oijen, 2017). Calibration is often applied to narrow the initial parameter ranges and capture regional or local peculiarities.

The challenges of model calibration include a selection of most influential variables to be calibrated using a sensitivity analysis (SA) and coping with equifinality, i.e. a situation when various combinations of parameter values produce the same results (Beven, 2006). To this end, different calibration approaches, such as trial-and-error parameter adjustments or probabilistic methods (Forrester et al., 2021; Hollós et al., 2022) including Bayesian methods (Fer et al., 2018; van Oijen, 2017) or the Generalized Likelihood Uncertainty Estimation (GLUE) (Beven and Binley, 2014) have been proposed. The calibration can focus on one or several variables simultaneously. Multivariate approaches are preferred as they aim to identify parameter values that minimise the differences between simulated and observed values of multiple variables (Kamali et al., 2022; Wöhling et al., 2013). The spatial aspect may be dealt with in a similar manner. The model calibrated for single sites provides outputs with a high local accuracy, whereas it loses the ability to generalize outside the calibration data (Blyth et al., 2011; Kramer et al., 2002; Levins, 1966). Therefore, introducing advanced calibration designs that are supposed to maintain a balance between the local accuracy and wide applicability, provide good performance across multiple variables, is needed.

In this study, we aimed to formulate a calibration workflow that offers improvements beyond the broadly used methods detailed in Keenan et al. (2011), and Wallach et al. (2021). We used the process-based model Biome-BGCMuSo (BBGCMuSo, Hidy et al., 2016, 2022, 2012), which is the most rapidly developing member of the Biome-BGC model family (Thornton, 1998). It simulates the storage and fluxes of carbon, nitrogen and water within and between the pools. The model has been extensively used in the research of forest ecosystems concerning their productivity (Kimball et al., 1997; Sever et al., 2017), carbon (Churkina et al., 2003; Ostrogović Sever et al., 2021; Yan et al., 2016), water (Pietsch et al., 2003), and nitrogen dynamics (Merganičová et al., 2005; Pietsch et al., 2003), including effects of climate change (Churkina and Running, 2000; Hlásny et al., 2011; Jager et al., 2000; White et al., 1999). The recent developments of BBGCMuSo included a multilayer soil profile; complex water cycling between soil, vegetation, and atmosphere; intra-annual phenology; and complex management operations (Hidy et al., 2012, 2016, 2022). However, robust testing of ecological plausibility and model performance in forest ecosystems as well as regionally calibrated species-specific parameter sets are still lacking. These tasks are challenging given the substantial increase in model structural complexity and the number of parameters, which limits the use of former parameter sets (Pietsch et al., 2005).

The aim of this study is to develop a multi-objective calibration procedure of model parameters that considers balancing the trade-off between the local precision of model outputs and a broad applicability of parameters, and to perform a comprehensive model benchmarking of the ecological plausibility of model results across large environmental gradients. We hypothesise that parameter estimates optimized for single sites are not sufficiently robust to be applied across large geographical space, while the multi-site optimisation reduces the ability of model to capture the phenotypic plasticity of vegetation that causes alterations in plant properties, e.g. allocation ratios between different plant organs in response to environmental conditions, thereby reducing the local accuracy (Gratani, 2014). The proposed method was applied to calibrate ecophysiological parameters of BBGCMuSo v6.2 for the European beech (*Fagus sylvatica* L.), the most widespread deciduous tree species in Europe. These findings may serve as a reference for calibrating other tree species and/or different models.

## 2. Data and Methods

### 2.1 Model

Biome-BGCMuso (Hidy et al., 2012, 2016, 2022) is a descendant of Biome-BGC model (Thornton, 1998). It is a biogeochemical model that simulates cycling of carbon, water, nitrogen, and energy in terrestrial ecosystems at a daily time step (Thornton, 1998). Biome-BGC was one of the earliest biogeochemical models that included explicit carbon, water and nutrient cycles. The represented processes include photosynthesis, evapotranspiration, allocation, respiration, litterfall, and decomposition (Thornton et al., 2002). These processes are defined for a unit ground area that is considered homogeneous. A so-called „two-leaf" model that represents stand foliage with one sunlit and one shaded leaf is used to simulate radiation interception, evapotranspiration, and gross primary production for the sunlit and shaded canopy fractions (Thornton and Rosenbloom 2005). The modelled ecosystem consists of several components representing different plant parts (leaf, stem, roots), litter, soil, and coarse woody debris. The ecosystem status and dynamics are represented by carbon (C), nitrogen (N) and water (W) pools and fluxes between the pools. The main pools represent leaf (C, W, N), aboveground wood (C, N), coarse root (C, N), fine root (C, N), coarse woody debris (C, N), litter (C, N), soil (C, W, N), yield (C, N), standing-dead-biomass (C, N) cut-down biomass (C, N). BBGCMuSo contains a number of new features including a multilayer soil representation that allows simulating more realistic dynamics of water, carbon and nitrogen across the soil profile; a possibility of using dynamic annual mortality rates; adjustable intra-annual allocation driven by phenology; improved representation of transpiration, soil evaporation, and inorganic nitrogen; and flexible simulation of management operations, including forest thinning and harvesting (Hidy et al., 2012, 2016, 2022). It can also simulate acclimation to temperature and short-term temperature dependence of maintenance respiration, drought legacy effects through reduction of non-structural carbohydrate storage pools, and a $CO_2$ concentration-dependent stomatal conductance (Hidy et al., 2016). The model version 6.2 uses 53 soil-related and 105 ecophysiological parameters (Hidy et al., 2021), of which some are site-specific (e.g. soil depth, soil texture), while others, such as C:N ratio in different tree compartments, are species-specific. The number of parameters in BBGCMuSo has been

tripled in comparison to the original model Biome-BGC, although 17 parameters concern crops and are not relevant for forest ecosystems. To perform simulations, a species or a plant functional type needs to be defined. In addition, site, soil and daily climate data are required inputs. In the case of forest ecosystems, stand age and past forest management are also necessary input data. The model, including its source code, is available at https://nimbus.elte.hu/bbgc/.

## 2.2 Data

The dendrometric and environmental data represented 87 forest sites distributed across Central Europe within the distributional range of European beech (Figure 1). The dataset includes sites from the International Co-operative Programme on Assessment and Monitoring of Air Pollution Effects on Forests (ICP Forests, Michel et al., 2018) long-term forest research plots from thinning trials supervised by different national institutions, and highly-instrumented intensively monitored plots equipped with weather stations, dendrometers, instruments measuring sap flow, soil water content, etc. (Table S2). The plots are located in

Croatia, Hungary, Slovakia, the Czech Republic, and Poland along an elevation gradient from 20 to 1,325 m a.s.l. Their mean annual precipitation totals vary from 419 to 1,883 mm, mean annual temperatures range from 3.5 to 13.3°C, and soil depths vary between 0.4 and 2 m (Table 1). Most plots were of a circular shape with a size from 0.09 to 1.05 ha. Forest stands were established between 1787 and 1984, i.e., their age in the year 2022 varied from 38 to 235 years. Both managed and unmanaged forest stands originating either from natural or artificial regeneration or a combination thereof, are represented. Time series

lengths and the number of observations differed between the sites depending on the year of plot establishment, and the frequency of re-measurements of tree dimensions. The maximum time series length was 60 years, and the maximum number of observations in a single series was 30.

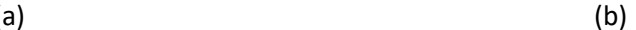

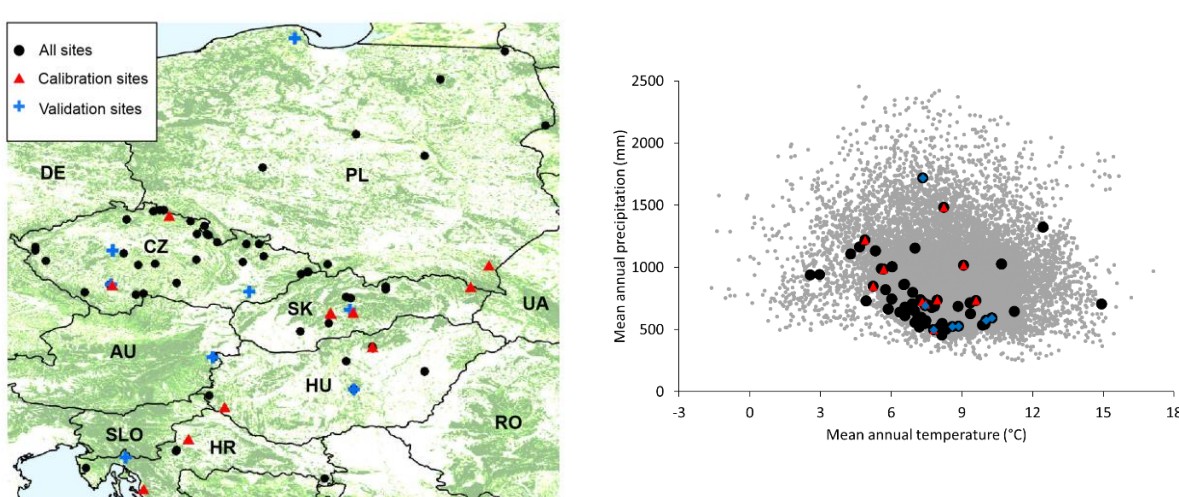

**Figure 1 a) Distribution of used forest sites across Europe with forest cover displayed in the background (CORINE Land Cover, 2023), b) The coverage of the entire climatic space of European beech in Europe (represented by grey dots using the data from Caudullo et al. 2017) by the used forest sites. Black dots indicate 87 sites used for testing model plausibility. Red triangles indicate eleven data-rich sites used for the calibration of the BBGCMuSo model. Blue crosses (a) and diamonds (b) represent eight validation sites.**

Out of the whole data set, 11 beech-dominated sites with the most comprehensive data and the balanced coverage of the geographical and environmental space were used for model calibration (hereafter referred to as calibration sites, Figure 1,

**Table 1**). The northern part of the selected region is underrepresented due to the insufficient data for model calibration at northern sites. Eight beech-dominated sites with repeated stand measurements covering the full range of environmental conditions represented in our database (

**Table 1**) were used for an independent model validation (hereafter referred to as validation sites). All 87 sites were used for testing the plausibility (realism) of simulated output. The comparison of beech natural distribution ranges (Pagan, 1996) with the covered ranges of latitude, elevation, climatic, and soil characteristics suggested that the selected sites should be representative of the Central European beech population.

**Table 1 Summary of site and forest stand characteristics for the whole data set (All), calibration and validation sets. Climate data represent a period from 1950 to 2018.**

| Data set | All | Calibration | Validation |
|---|---|---|---|
| Number of sites | 87 | 11 | 8 |
| Dominant tree species | *Fagus sylvatica, Picea abies, Pinus sylvestris, Quercus spp.* | *Fagus sylvatica* | *Fagus sylvatica* |
| **Site variables** | mean (min - max) | | |
| Years with observations | (1990 - 2018) | (1949 - 2018) | (1990 - 2018) |
| Latitude (°) | (N 45.4814 - N 54.559) | (N 44.8164 - N 50.7349) | (N 45.4814 - N 54.5592) |
| Longitude (°) | (E 14.2736 - E 23.72) | (E 14.3000 - E 22.4917) | (E 14.2736 - E 19.4701) |
| Elevation (m a.s.l.) | 544.2 (20 - 1325) | 686.8 (240 - 1325) | 502.5 (120 - 1180) |
| Annual precipitation (mm) | 789 (419 – 1884) | 1018 (663 – 1884) | 784 (458 - 1592) |
| Mean temperature (°C) | 7.75 (3.52 - 13.28) | 7.34 (4.86 - 10.3) | 8.37 (6.41 - 11.11) |
| Soil depth (m) | 0.99 (0.4 - 2) | 0.92 (0.45 - 2) | 0.99 (0.5 - 1.8) |
| Plot size (ha) | 0.31 (0.09 - 1.05) | 0.45 (0.25 - 1.05) | 0.28 (0.09 - 0.84) |
| Stand age (yr) | 84 (35 - 232) | 118 (64 - 232) | 79 (41 - 122) |
| Aboveground wood carbon (kgC m$^{-2}$) | 8.62 (0.01 - 36.87) | 16.92 (0.03 - 35.87) | 16.89 (4.30 - 33.74) |
| Soil carbon (kgC m$^{-2}$) | 13.31 (1.13 - 16.24) | 13.21 (1.13 - 16.24) | NA |
| Litter carbon (kgC m$^{-2}$) | 0.256 (0.086 - 0.344) | 0.256 (0.086 - 0.344) | NA |

The dataset contains information on site, forest stand structure and development, soil, climate, nitrogen deposition, and physiological processes (Table S2). Site description data comprise plot coordinates (latitude, longitude), elevation, aspect, and slope. Forest stand data comprise information on tree species composition, stand age, stand structure, individual tree

dimensions or mean stand characteristics (e.g. mean diameter at breast height (DBH), mean tree height (H)), mortality, the applied management and damages. Soil dataset contains soil depth, texture, pH, nutrient stocks, and indicators of water regime. If soil information was not available, it was obtained from the Harmonized World Soil Database (HWSD v 1.21, FAO, 2012) that provides soil attributes, such as soil depth, soil texture, pH, etc., at a grid cell size of approximately 1km.

Climate data include daily values of minimum and maximum temperature, solar radiation, precipitation, and vapour pressure deficit (VPD). These data were compiled from different sources, including observations at individual sites, nearby meteorological stations, or if no local data were available, the E-OBS gridded dataset providing daily minimum, maximum temperature, and precipitation with 0.1 deg. resolution was used (Cornes et al., 2018). The climate data covered a period from 1950 to 2018. To cope with limited availability of the local data, we combined the on-site measurements with the E-OBS data. The MTClim model (Hungerford et al., 1989) was used to extrapolate the climate time series from the nearest E-OBS grid cell to account for the elevation difference between the source cell and the target site, and to calculate daylight values of mean temperature, VPD, solar radiation, and daylength at individual sites required by BBGCMuSo.

Annual $CO_2$ data were taken from Mauna Loa observations (Keeling et al., 1976). Since nitrogen deposition data were directly available only for ICP Forests plots, they were taken from Tian et al. (2018) for the remaining sites.

Data about site, soil texture, pH, stand age, management, nitrogen deposition, $CO_2$ concentration and daily climate data were used as model inputs to run site-specific simulations. Model calibration and validation was based on carbon stocks in aboveground wood, litter and soil (AbgwC, SoilC, LitterC). AbgwC was derived from the dendrometric characteristics of individual trees by calculating first the total aboveground wood volume using species-specific two-parameter regressions (Petráš and Pajtík 1991). The produced values were subsequently converted to carbon stock using the species-specific basic wood density (Merganič et al. 2017, 570 kg m$^{-3}$ for beech) and 50 % carbon content in biomass (IPCC, 2003).

## 2.3 Simulation design

The simulations were performed in three steps: (1) spin-up run, (2) transient run, and (3) normal run (Hidy et al., 2021). The spin-up was performed with a constant $CO_2$ concentration and nitrogen deposition equal to the pre-industrial values of 277.15 ppm and 0.002 kgN m$^{-2}$ year$^{-1}$, respectively. During the transient run, $CO_2$ concentration and nitrogen deposition increased annually from the pre-industrial to the current values. The transient run started in 1850 and lasted until the year preceding the establishment of the current stand. Hence, the length of the transient run varied between sites, and it was used only for the simulations of the stands established after 1850, while the maximum length of the transient run was 134 years. Both spin-up and transient runs were performed with no management, no fire-induced mortality, and constant natural (background) mortality.

The normal run was driven by the temporally varying $CO_2$ and N-deposition, and included management reconstructed based on forest inventory records or yield tables, if no site-specific information on management interventions was available. The simulations started at stand age 0 (i.e., the year of stand establishment) and continued to the present day; i.e. the simulation

length equalled the actual stand age. In simulation year 1, a clearcut was applied followed by the removal of 90 % of the aboveground woody biomass accumulated during the spin-up and transient runs, with all non-woody biomass, i.e. the foliage, remaining in the stand. Natural mortality was changing annually (Fig. S1). In the first 30 years after the stand establishment, natural mortality rates followed a decreasing exponential function reaching the highest annual mortality rate one year after the stand establishment and its subsequent gradual reduction over time, resembling the survival rates of forest regeneration from experimental studies focusing on beech (Barna et al., 2011; Hülsmann et al., 2018). After 30 years, we used a constant annual mortality rate of 0.9 % (Pajtík et al. 2018; Vanoni et al. 2019) in managed stands, while unmanaged stands were simulated using the dynamic natural mortality rates that fluctuated between 0.76 and 4.1 % during a cycle of 300 years, following the elliptical function (Merganičová and Merganič, 2014).

## 2.4 Parameter estimation

The estimation of model parameters (i.e. model calibration) consisted of several phases (Figure 2) described below.

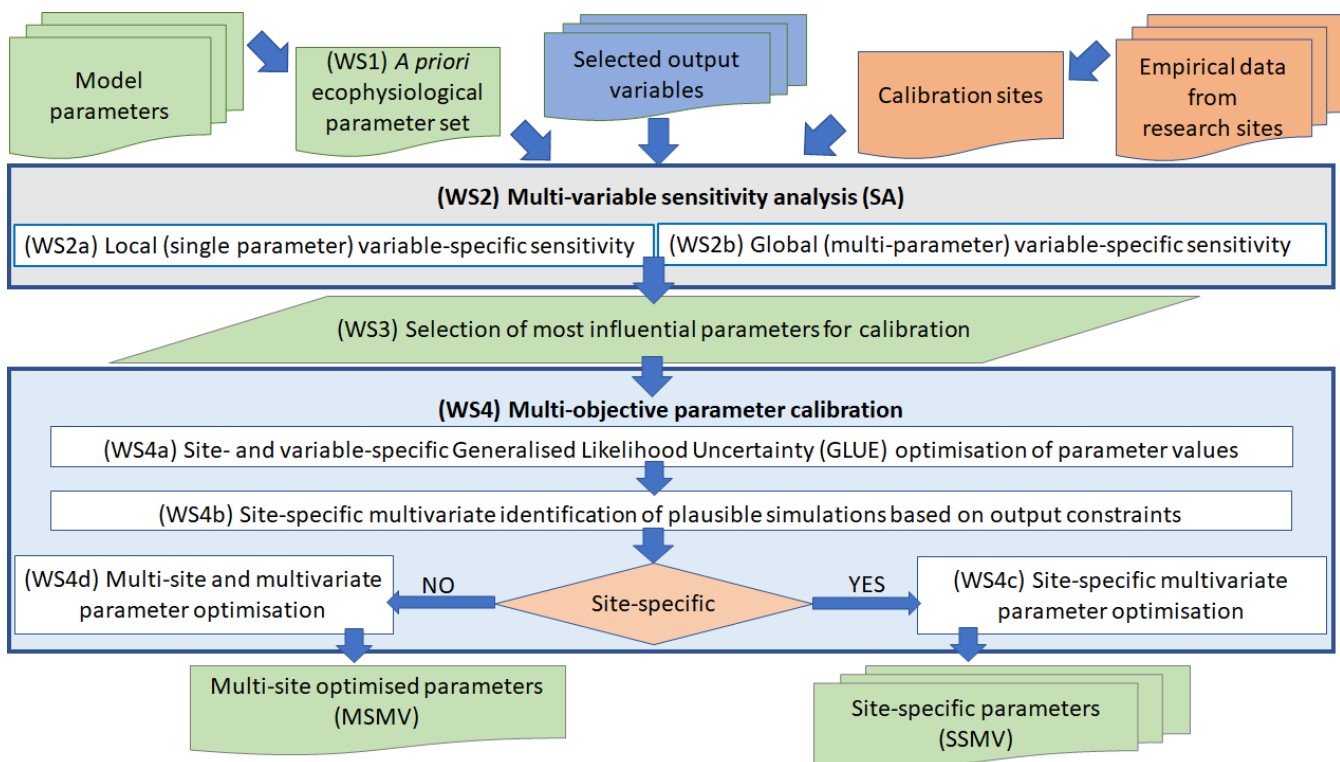

**Figure 2 A general workflow used for the optimisation of ecophysiological parameters of BBGCMuSo for the European beech using the multi-objective calibration approach.**

### 2.4.1 A priori parameter setting

Prior to the calibration of model parameters, the default or a priori parameter values were set (the WS1 phase in Figure 2). The initially defined set (column "First" in Table 2) was defined based on values of 40 ecophysiological parameters included in previous model versions taken from Pietsch et al. (2005). The values of new 48 parameters, which were introduced in the latest
model version, were set based on the literature review and the TRY database (Kattge et al., 2020). This set was used for the first simulations of all 87 sites to examine the success of simulated development. By a successful simulation we understand a simulation from spin-up until the end of the normal run, during which the ecosystem stability was maintained, i.e. carbon stocks in vegetation and soil were accumulated during the spin-up until they reached a balanced state, and the vegetation existence, confirmed by non-zero values of vegetation (AbgwC, foliage, roots) carbon pools, was maintained throughout the
entire normal run simulation. Based on the results of unsuccessful simulations, we identified parameters, the values of which may have caused the problems, such as insufficient water or nitrogen supply resulting in the cessation of vegetation existence. Subsequently, we used a trial-and-error approach to alter the respective parameters until we reached a 100 % success rate of the simulations. Those parameter values represented an a priori parameter set (column "A priori" in Table 2).

**Table 2 List of parameters tested and modified during the calibration of Biome-BGCMuSo for *Fagus sylvatica* L. Columns Min and Max represent the minimum and maximum values used to define the parameter ranges for the sensitivity analysis and optimisation. Local Sensitivity contains the values of the sensitivity index per output variable (carbon stock in aboveground wood, soil, and litter labelled as AbgwC, SoilC, and LitterC) calculated following Hoffman and Gardner (1983, Eq. 1). The column "First" is the first parameter approximation identified based on the species-specific values derived for Biome-BGC by Pietsch et al. (2005) (in italics)**
**and the complementary literature review. The column "A priori" contains parameter values that enabled the successful simulations of all 87 sites(result of the WS1 phase in Figure 2). The column "MSMV" contains the multi-site and multivariate calibrated parameter set (result of the WS4d phase in Figure 2) based on 11 calibration sites and 3 output variables (AbgwC, SoilC, and LitterC). The calibrated parameters are indicated with the grey background.**

| Parameter | | | Tested range | | Local sensitivity | | | Parameter set | | |
| Name | Abbreviation | Unit | Min | Max | AbgwC | SoilC | LitterC | First | A priori | MSMV |
|---|---|---|---|---|---|---|---|---|---|---|
| Transfer growth period as fraction of growing season | GP | dim | 0.05 | 0.3 | 0.005 | 0.073 | 0.003 | 0.2 | 0.2 | 0.2 |
| Litterfall as fraction of growing season | LP | dim | 0.2 | 0.6 | 0.002 | 0.027 | 0.001 | 0.2 | 0.2 | 0.2 |
| Base temperature | T_base | °C | 0 | 7 | 0 | 0 | 0 | 5 | 5 | 5 |
| Annual live wood turnover fraction | WTF | dim | 0.5 | 1 | 0.00002 | 0.00026 | 0.00001 | 0.7 | 0.7 | 0.7 |
| Annual fire mortality fraction | FM | dim | 0 | 1 | 0.078 | 0.716 | 0.008 | 0 | 0 | 0 |
| Whole-plant mortality fraction in vegetation period | WPM | dim | 0 | 0.1 | 0.996 | 0.493 | 0.006 | 0.005 | 0.005 | 0.005 |
| C:N of leaves | CN_lv | kgC kgN⁻¹ | 16.5 | 40 | 0.017 | 0.183 | 0.004 | 26.9 | 26.9 | 26.9 |
| C:N of leaf litter, after retranslocation | CN_li | kgC kgN⁻¹ | 10 | 114 | 0.0002 | 0.015 | 0.178 | 44 | 44 | 44 |
| C:N of fine roots | CN_ro | kgC kgN⁻¹ | 10 | 75.8 | 0.005 | 0.054 | 0.007 | 47.6 | 47.6 | 47.6 |
| C:N of live wood | CN_lw | kgC kgN⁻¹ | 17 | 100 | 0.004 | 0.041 | 0.001 | 50 | 50 | 50 |
| C:N of dead wood | CN_dw | kgC kgN⁻¹ | 300 | 819 | 0.00002 | 0.0002 | 0.005 | 550 | 550 | 550 |
| Leaf litter labile proportion | LLaP | dim | 0.1 | 0.6 | 0.00002 | 0.111 | 0.004 | 0.124 | 0.124 | 0.124 |
| Leaf litter cellulose proportion | LCeP | dim | 0.1 | 0.7 | 0.00002 | 0.157 | 0.004 | 0.561 | 0.561 | 0.561 |
| Fine root labile proportion | RLaP | dim | 0.1 | 0.6 | 0.00002 | 0.067 | 0.003 | 0.34 | 0.34 | 0.34 |
| Fine root cellulose proportion | RCeP | dim | 0.1 | 0.6 | 0.00003 | 0.080 | 0.002 | 0.44 | 0.44 | 0.44 |
| Dead wood cellulose proportion | WCeP | dim | 0.5 | 0.9 | 0.00004 | 0.361 | 0.376 | 0.77 | 0.77 | 0.77 |
| Canopy water interception coefficient | CWIC | mm LAI⁻¹ d⁻¹ | 0.01 | 0.063 | 0.003 | 0.017 | 0.011 | 0.034 | 0.034 | 0.034 |
| Canopy light extinction coefficient | CLEC | dim | 0.3 | 0.7 | 0.012 | 0.136 | 0.004 | 0.6 | 0.6 | 0.6616 |
| All-sided to projected leaf area ratio | SLA:PA | dim | 1.5 | 2.5 | 0.0007 | 0.001 | 0.00002 | 2 | 2 | 2 |
| Ratio of shaded SLA:sunlit SLA | shSLA:suSLA | dim | 0.2 | 5 | 0.073 | 0.568 | 0.004 | 2 | 2 | 2 |
| Fraction of leaf N in Rubisco | FLNR | dim | 0.1 | 0.3 | 0.008 | 0.087 | 0.003 | 0.162 | 0.162 | 0.1383 |
| Maximum stomatal conductance (projected area basis) | MSC | m s⁻¹ | 0.001 | 0.009 | 0.088 | 0.925 | 0.020 | 0.006 | 0.005 | 0.0051 |
| Cuticular conductance (projected area basis) | CC | m s⁻¹ | 0.00001 | 0.0001 | 0.001 | 0.011 | 0.000 | 0.00006 | 0.00006 | 0.00006 |
| Boundary layer conductance (projected area basis) | BLC | m s⁻¹ | 0.01 | 0.09 | 0.005 | 0.076 | 0.002 | 0.01 | 0.01 | 0.01 |
| Maximum depth of rooting zone | MRD | m | 0.2 | 4.1 | 0.001 | 0.021 | 0.003 | 2 | 2 | 2 |
| Root distribution parameter | rootDistr | dim | 0.5 | 4 | 0.001 | 0.088 | 0.123 | 3.67 | 1.5 | 1.5 |
| Growth resp per unit of C grown | GRC | dim | 0.1 | 0.5 | 0.020 | 0.216 | 0.016 | 0.3 | 0.3 | 0.3 |
| Maintenance respiration in kgC/day per kg of tissue N | MRperN | kgC kgN⁻¹ d⁻¹ | 0.1 | 0.4 | 0.091 | 0.902 | 0.023 | 0.218 | 0.218 | 0.218 |
| Theoretical maximum proportion of non-structural and structural carbohydrates | NSC:SCmax | dim | 0.05 | 0.3 | 0.0001 | 0.0004 | 0.00001 | 0.1 | 0.1 | 0.1 |

| Description | Abbreviation | unit | | | | | | | | |
|---|---|---|---|---|---|---|---|---|---|---|
| Proportion of non-structural carbohydrates available for maintenance respiration | NSC2MR | dim | 0.1 | 0.5 | 0.00002 | 0.00011 | 0.000008 | *0.3* | 0.3 | 0.3 |
| Symbiotic+asymbiotic fixation of N | Nfix | kgN m⁻² yr⁻¹ | 0.0001 | 0.01 | 0.046 | 0.494 | 0.991 | *0.0005* | 0.01 | 0.0091 |
| Time delay for temperature in photosynthesis acclimation | tau | day | 0 | 50 | 0.0003 | 0.0035 | 0.0001 | *10* | 10 | 10 |
| Volumetric water content ratio to calculate soil moisture limit 1 | VWCratio_lim1 | dim | 0.1 | 0.9 | 0.004 | 0.067 | 0.002 | *0.99* | 0.1 | 0.1 |
| Volumetric water content ratio to calculate soil moisture limit 2 | VWCratio_limit2 | dim | 0.5 | 1 | 0.0001 | 0.0049 | 0.0021 | *0.99* | 0.99 | 0.99 |
| Minimum of soil moisture limit2 multiplicator (full anoxic stress value) | min_soilstress2 | dim | 0 | 1 | 0 | 0 | 0 | *0.4* | 0.4 | 0.4 |
| Vapor pressure deficit: start of conductance reduction | VPDS | Pa | 500 | 1500 | 0.005 | 0.079 | 0.003 | *600* | 600 | 600 |
| Vapor pressure deficit: complete conductance reduction | VPDC | Pa | 1500 | 3500 | 0.080 | 0.783 | 0.009 | *3000* | 3000 | 2910 |
| Maximum senescence mortality coefficient of aboveground plant material | SMCA | dim | 0 | 0.01 | 0 | 0 | 0 | *0.001* | 0 | 0 |
| Maximum senescence mortality coefficient of belowground plant material | SMCB | dim | 0 | 0.01 | 0.0001 | 0.001 | 0.00002 | *0.001* | 0 | 0 |
| Maximum senescence mortality coefficient of non-structured plant material | SMCL | dim | 0 | 0.01 | 0.003 | 0.079 | 0.004 | *0.0001* | 0 | 0 |
| Lower limit extreme high temperature effect on senescence mortality | SNSC_ext1 | °C | 30 | 40 | 0 | 0 | 0 | *30* | 30 | 30 |
| Upper limit extreme high temperature effect on senescence mortality | SNSC_ext2 | °C | 30 | 50 | 0 | 0 | 0 | *40* | 40 | 40 |
| Turnover rate of wilted standing biomass to litter | TRWB | dim | 0.01 | 0.1 | 0 | 0 | 0 | *0.01* | 0.01 | 0.01 |
| Turnover rate of non-woody cut-down biomass to litter | TRCN | dim | 0.01 | 0.1 | 0 | 0 | 0 | *0.05* | 0.01 | 0.01 |
| Turnover rate of woody cut-down biomass to litter | TRCW | dim | 0.0001 | 0.1 | 0 | 0 | 0 | *0.01* | 0.0009 | 0.0009 |
| Drought tolerance parameter | DSWScirt | nday | 0 | 100 | 0 | 0 | 0 | *30* | 100 | 100 |
| Effect of soilstress factor on photosynthesis | Sseff | dim | 0 | 0.4 | 0.006 | 0.066 | 0.002 | 0 | 0 | 0 |
| Leaf carbon allocation proportion | | dim | - | - | - | - | - | *0.173* | 0.173 | 0.173 |
| Fine root carbon allocation proportion | | dim | - | - | - | - | - | *0.094* | 0.094 | 0.094 |
| Live woody stem carbon allocation proportion | | dim | - | - | - | - | - | *0.101* | 0.101 | 0.101 |
| Dead woody stem carbon allocation proportion | | dim | - | - | - | - | - | *0.556* | 0.556 | 0.556 |
| Live coarse root carbon allocation proportion | | dim | - | - | - | - | - | *0.012* | 0.012 | 0.012 |
| Dead coarse root carbon allocation proportion | | dim | - | - | - | - | - | *0.064* | 0.064 | 0.064 |
| Canopy average specific leaf area (projected area basis) | | m2 kgC⁻¹ | - | - | - | - | - | *48* | 34.5 | 34.5 |
| Canopy growth proportion | | dim | - | - | - | - | - | *0.5* | 0.5 | 0.5 |

## 2.4.2. Sensitivity analysis

The sensitivity analysis (SA, the WS2 phase in Figure 2) was performed to identify the effect of model parameters on simulated carbon stocks in aboveground wood, soil, and litter (AbgwC, SoilC, LitterC) at the calibration sites (

**Table 1**). The three carbon stock variables were selected instead of typically used carbon fluxes to cover a wider range of environmental conditions, since fluxes are usually measured only at a limited number of sites. Moreover, BBGCMuso has already been found to effectively simulate C fluxes (Hidy et al., 2016; Maselli et al., 2009), while its ability to simulate C stocks is much less documented (Ostrogović Sever et al., 2021). Moreover, focusing on C stocks rather than fluxes enables us to evaluate model performance over the multidecadal time span of up to 69 years represented by our data.

First, we performed a local, i.e. single parameter, SA (WS2a in Figure 2) using regular sampling of parameter values from their pre-defined ranges based on the literature review. The sensitivity of variable $i$ to parameter $P$ was quantified using the sensitivity index ($SI$) proposed by Hoffman and Gardner (1983):

$$SI_{Pi} = \frac{Vmax_{Pi} - Vmin_{Pi}}{Vmax_i} \qquad (1)$$

where $Vmax_{Pi}$ and $Vmin_{Pi}$ are the maximum and minimum values of the simulated output variable $i$ when testing the effect of parameter $P$, and $Vmax_i$ is the absolute maximum value of the output variable obtained from the tests of all parameters.

Afterwards, a global, i.e. multi-parameter, SA (WS2b in Figure 2), which assesses the sensitivity of all selected parameters across the entire parameter space simultaneously, was performed using the least squares linearisation (LSL) approach (Verbeeck et al., 2006). A commonly used global SA method is the variance-based Sobol analysis, which performs Monte-Carlo simulations on the parameter-space. The method estimates the Sobol sensitivity index that distributes the overall variability of model outputs to the contributions from each model input (Saltelli et al., 2004). As the parameter space expands, an increasing number of Monte-Carlo simulations is required to accurately estimate the sensitivity. To simplify this process and enhance the accuracy with a fewer number of simulations, surrogate models are employed. The simplest surrogate models are multivariable linear models (Verbeeck et al. 2006), including LSL. This approach utilises the ordinary least squares method to approximate the process-based model with a surrogate multidimensional linear model. The coefficients derived from the fitted model are then used to calculate the relative Sobol sensitivity indices.

The procedure first simultaneously samples values of all selected model parameters from their predefined ranges with Monte Carlo simulations, while assuming a multivariate uniform distribution. Then, the simulated outputs are examined with regard to parameter deviations from the mean using the LSL, which splits the overall output uncertainty into its individual sources. This allowed us to estimate the contribution of each tested parameter to the model output uncertainty and identify the parameters that affected AbgwC, SoilC, and LitterC most. The results of SA were used to select the parameters to be calibrated (WS3 in Figure 2). The SA was performed using the musoMonte and musoSensi functions implemented in the RBBGCMuSo package (Hollós et al., 2023) available at https://github.com/hollorol/RBBGCMuso.

### 2.4.3 Parameter calibration

First, we performed a site- and variable-specific calibration (WS4a in Figure 2) using the GLUE method (Beven and Binley, 2014) implemented in the calibMuso function of the RBBGCMuSo package (Hollós et al., 2023). With this procedure, a selected parameter set was optimised using the least squares likelihood function based on the comparison of the simulated values of the selected output variable with the corresponding observations in the pre-defined parameter space:

$$L = e^{-\sqrt{\frac{(Vobs-Vsim)^2}{n}}} \tag{2}$$

where $L$ is the estimated likelihood, *Vobs* and *Vsim* are the observed and the simulated values of the output variable, and $n$ is the number of observations.

For every site, we performed 100,000 Monte Carlo simulations, each with a unique combination of parameter values randomly sampled from the predefined parameter ranges (Table 2).

Then, we evaluated the plausibility of simulated AbgwC, LitterC, and SoilC (WS4b in Figure 2) at the end of the spin-up and in individual years of the normal run simulations. We applied the following constraints derived from the literature to evaluate the plausibility of simulated values: carbon stock in aboveground wood below 70 kgC m$^{-2}$ (Barna et al., 2011; Georgi et al.,

2018; Standovár and Kenderes, 2003; Trotsiuk et al., 2012), soil carbon in the whole soil profile between 5 and 25 kgC m$^{-2}$, and litter carbon amount between 0.1 to and 4 kgC m$^{-2}$ (De Vos and Cools, 2011; Pavlenda and Pajtík, 2010; Wellbrock et al., 2016; Wellbrock and Bolte, 2019). A simulation was identified as plausible if all three examined output variables were within the given ranges.

The site-specific multivariate optimised parameter values (SSMV parameter sets resulting from the WS4c phase in Figure 2)
were derived from the subsets of plausible simulations selected for each calibration site as those minimising the estimation errors of AbgwC, SoilC, and LitterC, and maximizing the joint-likelihood function (WS4b in Figure 2). We applied the normal likelihood function to each of the three output variables and afterwards calculated the sum of loglikelihood values for each year, assuming the independence of the estimation errors.

Next, we performed a parameter optimisation across the studied geographical domain (WS4d in Figure 2) by processing the
315 plausible simulations from calibration sites (selected in the WS4b phase in Figure 2). We identified feasible ranges for each parameter and site based on intersections of variable-specific parameter ranges (Figure 3a).Then, we derived the multi-site feasible parameter ranges by intersecting the site-specific feasible parameter ranges (Figure 3b) in the tested 6D parameter space defined by the calibrated parameters. Afterwards, we divided the multi-site feasible ranges of each parameter into five equally wide sub-intervals that define 5 discrete steps on each dimension (Figure 3b) of the multi-dimensional parameter space.
This categorisation was needed because the applied parameter values in site-specific Monte Carlo simulations differed between the sites. In the hyperspace, we identified ten cells with the highest number of allocated sites and simulations and the smallest arithmetic mean errors of AbgwC, SoilC, and LitterC. Finally, we calculated mean parameter values for each cell. The final multi-objective, i.e. multi-site multivariate (MSMV) optimal parameter set was the one that led to successful simulations of all calibration sites and generated the smallest errors.

(a)

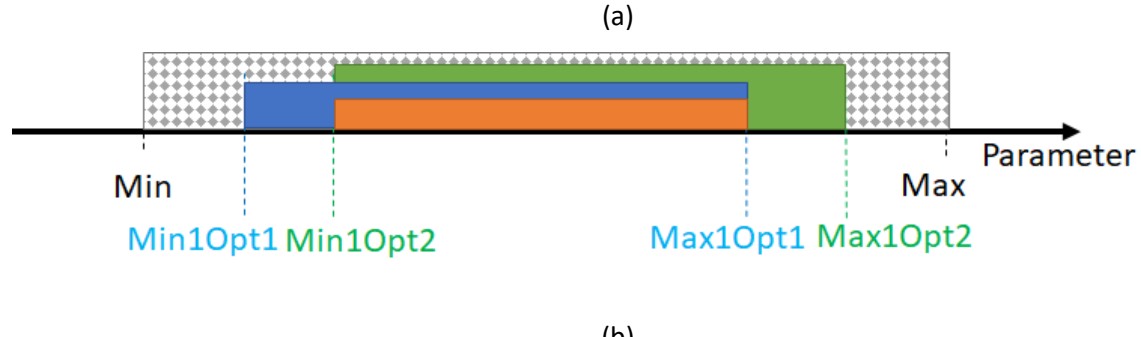

(b)

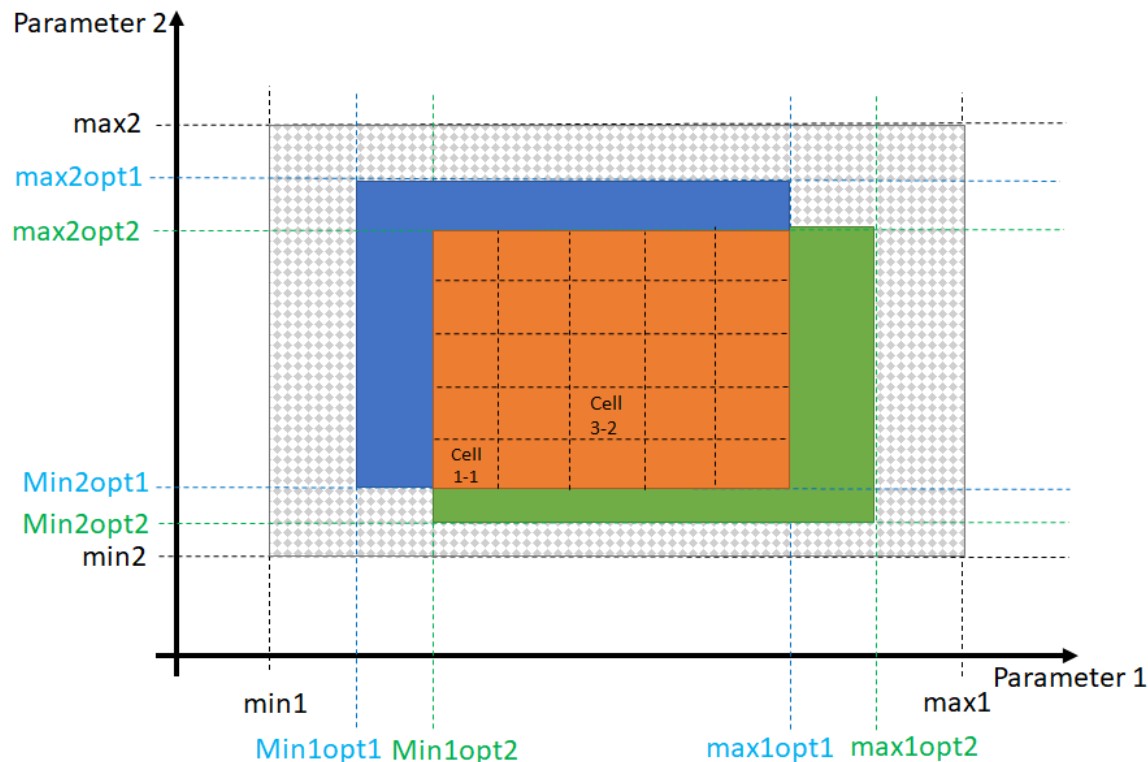

**Figure 3** Identification of site-specific feasible parameter ranges using a multivariate approach (a); and identification of multi-site and multivariate optimised (MSMV) parameter values and their parameter ranges (b). The approach in (a) is demonstrated using an example for a single site, a single parameter and two variables. The tested parameter interval is indicated by the grey dotted pattern, feasible parameter ranges for individual variables are indicated with blue and green colours, and the multivariate feasible parameter range if shown in orange. The approach (b) is demonstrated using a hypothetical 2D parameter space and site-specific optimised parameter ranges for two sites. The grey dotted space indicates the tested 2D parameter space, the blue and green colours refer to 2D parameter space for specific sites, and the orange space refers to multi-site and multivariate feasible parameter ranges. Min = minimum, Max = maximum, Opt = optimised values of tested parameters.

## 2.5 Robustness, validation and plausibility tests

### 2.5.1 Robustness of calibrated parameter values

The robustness of the site-specific (SSMV) and multi-site (MSMV) optimised values of parameters was tested by simulating all calibration sites with all derived parameter sets and calculating the root mean square errors (RMSE) of the simulated output variables. This test allowed us to examine the possible overfitting as well as the applicability of site-specific parameter sets outside the actual site conditions.

Next, we applied decision trees (DT) to identify the problems arising from the trade-off between variables and/or sites based on the evaluation of the entropy. For each site and output variable we determined parameter ranges, within which the plausible simulations can be expected with the highest probability by constructing DT from the outputs of 100,000 Monte Carlo simulations per calibration site. The simulations were split into plausible and implausible ones based on the constraints

specified for the three tested variables (done under the WS4b phase in Fig. 2, Sect. 2.4.3.). In total we derived 33 decision trees (11 sites x 3 variables) using the rpart R package (Therneau et al., 2023). The plausible ranges for each parameter, site and

variable were determined based on the selection of the most probable leaf node. The ranges may differ from those obtained under the WS4b phase, where each parameter was analysed separately (Figure 3a), because DT evaluate all parameters at the same time (Figure 4). Then, we searched for the intersections of these ranges to obtain the final parameter-wise plausible ranges for a specific site and all variables together and for multi-site ranges for all calibration sites and variables together (Figure 4). The SSMV and MSMV optimised parameter values were considered robust if they occurred within the respective

DT parameter ranges.

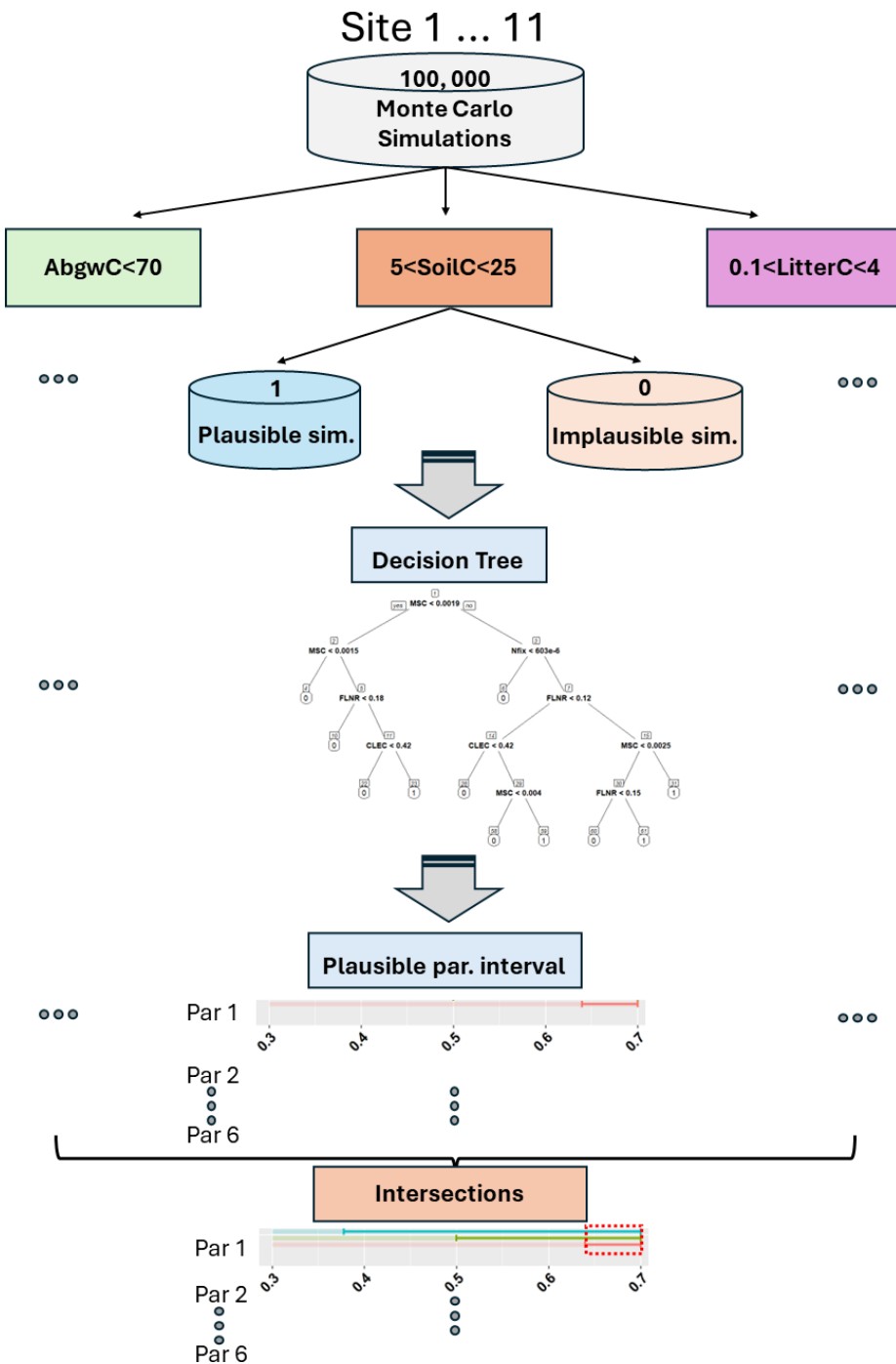

**Figure 4 Determination of plausible ranges of parameters using decision trees across all sites and output variables. Par = parameter, background transparent coloured horizontal lines represent the tested parameter range, while non-transparent coloured lines show the optimised parameter ranges based on decision trees performed for carbon stock in aboveground wood (AbgwC), soil (SoilC), and litter (LitterC). The dashed red rectangle represents the plausible range of a single parameter based on the intersection of its plausible ranges derived for the three output variables.**

The next test was aimed at analysing if SSMV parameter values followed any trends along specific gradients. Specifically, we examined the interdependencies between the pairs of parameters, and the trends of their site-specific values along stand and site gradients, namely age, elevation, latitude, climate and soil characteristics. Any significant trend may indicate that the respective parameter should not be handled as a constant but should vary with specific conditions. The analyses were based on Spearman correlations, and linear regressions performed in R environment (R Core Team, 2018). Subsequently, we examined the physiological meaningfulness of revealed trends and relationships using the empirical evidence collected from the scientific literature.

Following the identification of the significant relationships and their biological plausibility, we derived ten multiple linear regression models explaining the variation in site-specific values of the canopy light extinction coefficient (CLEC) of calibration sites using different combinations of environmental predictors and analysed the explanatory power and performance of the models with the following statistical characteristics: R-squared, adjusted R-squared, AKAIKE information criterion, Bayesian information criterion, Mallows' statistic, and the residual standard deviation. This was performed in R environment using the car (Fox et al., 2023) and lmSubsets (Hofmann et al., 2021) R packages. Then we applied the derived functions to all 87 sites, for which we calculated site-specific values of CLEC. and simulated each site with the respective CLEC value. Afterwards, we examined the robustness and plausibility of the simulated output with varying CLEC across the whole geographical domain.

### 2.5.2 Model performance with parameter values optimised for multiple sites

The MSMV parameter set derived in the WS4d calibration phase (Figure 2) was validated using an independent dataset from 8 European beech-dominated sites (Table 1, Figure 1), each represented by at least two repeated observations of the aboveground wood carbon. The simulations consisted of spin-up, transient run and normal run, as described in Sect. 2.3. The validation was based on the comparison of modelled and observed carbon stocks in aboveground wood at specific time points. We calculated the bias defined as an arithmetic mean of differences between modelled and observed values of the respective variable, mean absolute error (MAE), mean percentage error (MPE), and root mean square error (RMSE). SoilC and LitterC were compared to the plausible ranges derived from the literature (De Vos and Cools, 2011; Pavlenda and Pajtík, 2010; Wellbrock et al., 2016; Wellbrock and Bolte, 2019), since no observed data on soil and litter carbon were available for the validation sites.

### 2.5.3 Robustness and plausibility of simulated output at a large scale

To evaluate the broad applicability of the derived multi-objective parameter set across the whole studied geographical region, we simulated forest development at 87 sites, encompassing the main climatic and soil gradients in the study area (Figure 1). We specifically assessed the plausibility of absolute values of AbgwC, LitterC, and SoilC by comparing them with values documented in the literature (e.g. Barna et al., 2011; Pavlenda and Pajtík, 2010). We also examined the responses of simulated carbon stocks to environmental conditions (e.g. latitude, longitude, elevation, annual precipitation, mean temperature, proportion of sand, silt, and clay fractions, soil depth) and compared the observed shapes to the patterns published in empirical studies. We analysed the responses of simulated outputs using Spearman correlations, linear and quadratic regressions, and generalised additive models (GAM). When explaining the patterns of the main three variables along environmental gradients we examined also other stocks, such as carbon stock in roots, leaves, as well as some carbon fluxes, particularly heterotrophic respiration, to reveal the mechanisms driving the model responses. All tests were performed in the R environment (R Core Team, 2018) and the results were visualised using ggplot2 R package (Wickham, 2016).

## 3. Results

### 3.1 Parameter sensitivity analysis

The local (single parameter) sensitivity analysis (WS2a in Figure 2) focusing on evaluating the effects of individual parameters revealed that the aboveground wood carbon stock was most affected by the whole plant mortality rate (WPM). Soil and litter carbon stocks were most sensitive to the maximum stomatal conductance (MSC) and nitrogen fixation (Nfix), respectively (Table 2). The analysis of trends in variable changes due to modifications of parameter values clarified how the increase of the parameter value affected the values of the respective output variable, e.g. the increase of MSC caused an increase of all tested output variables (AbgwC, SoilC, LitterC) in the whole parameter range (Fig. S4). The impact of other parameters was more complex, as we revealed both positive and negative trends, while in the case of the increase of e.g. Nfix the positive ones prevailed in AbgwC and SoilC, and negative ones in LitterC. A more detailed analysis identified the changes of output variables along the parameter range, e.g. the increase of Nfix caused an initial increase of LitterC, which was followed by its gradual reduction as Nfix was increasing (Fig. S2a).

The global (multi-parameter) sensitivity analysis (WS2b in Figure 2) showed that Nfix had the highest impact on all three analysed carbon pools (Figure 5). The subsequent parameters were MSC, growth respiration per unit of carbon allocation (GRC), maintenance respiration in kgC day$^{-1}$ per kg of tissue nitrogen (MRperN), fraction of leaf nitrogen in Rubisco (FLNR). However, the ranking of parameters differed between the individual output variables (Figure 5).

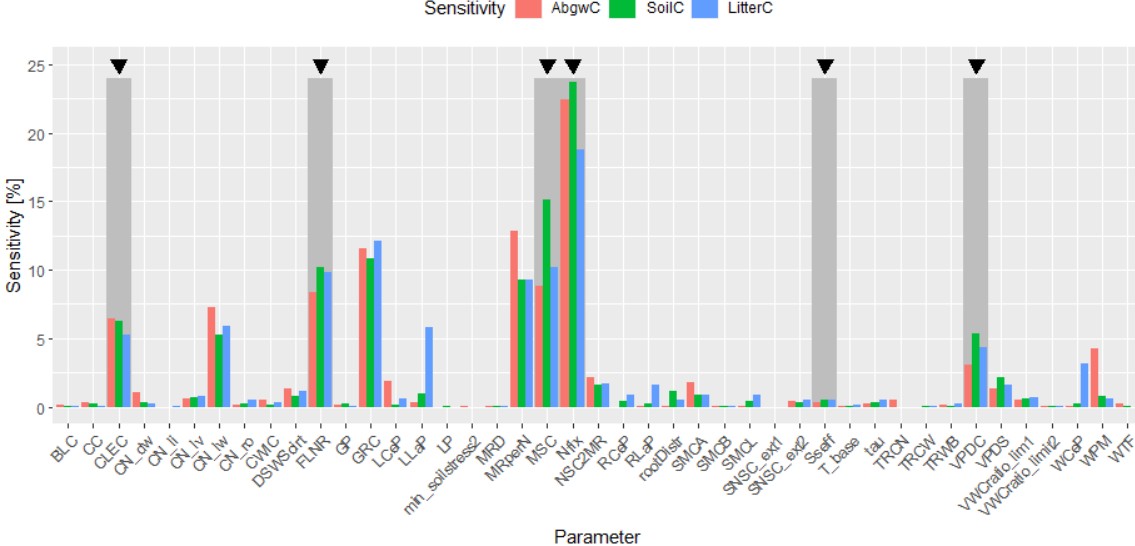

**Figure 5 Results of global (multi-parameter) sensitivity analysis (WS2b in Figure 2) of simulated carbon stock in aboveground wood, soil and litter (AbgwC, SoilC, LitterC) to ecophysiological parameters of BBGCMuSo. Parameter abbreviations are given in Table 2. Dark grey rectangles with black arrows above indicate parameters selected for the calibration procedure: canopy light extinction coefficient (CLEC), fraction of leaf nitrogen in rubisco (FLNR), maximum stomatal conductance (MSC), nitrogen fixation (Nfix),**

**effect of soil stress factor on photosynthesis (Sseff), and vapour pressure deficit for complete conductance reduction (VPDC).**

Based on the results of SA, we selected six parameters to be calibrated: canopy light extinction coefficient (CLEC), FLNR, MSC, Nfix, vapour pressure deficit for complete conductance reduction (VPDC), and effect of soil stress factor on photosynthesis (Sseff), as they had a substantial effect on carbon stock in aboveground wood, soil, and litter (Figure 5). Other parameters with a high influence on the simulated C pools (such as C:N ratio in leaves, Fig. S2b) were not to selected for

calibration due to a strong support of their actual values from the literature (e.g. Fig. S3).

### 3.2 Parameter estimation

The site-specific multivariate (SSMV) optimised values of all 6 calibrated parameters for 11 calibration sites differed from the a priori ones, varied within the whole tested ranges (Table 2) and strongly differed between the sites (Figure 6). Two parameters (FLNR, VPDC) showed a gradual change of SSMV values across the whole tested range, while the SSMV values of others,

especially CLEC and Sseff, were clustered. In comparison to a priori values, SSMV values of individual parameters changed by 41 % on average, while Nfix was modified most substantially (67 % on average).

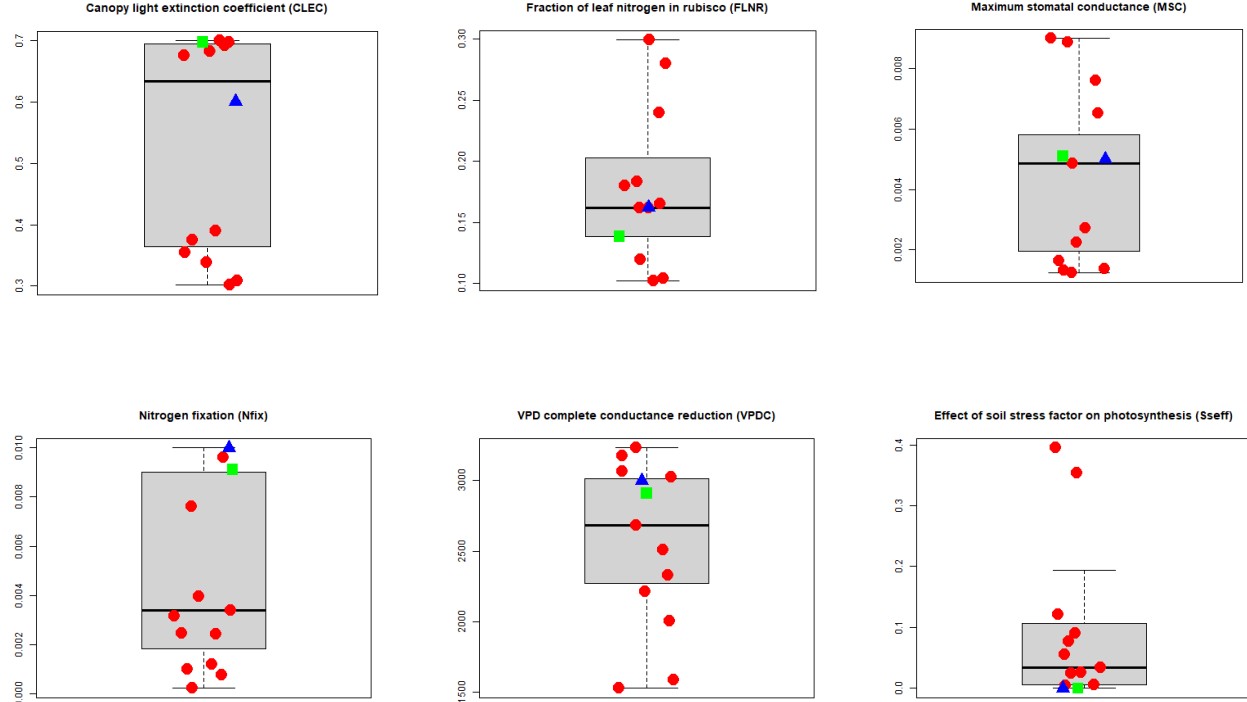

**Figure 6 A priori (blue triangles), site-specific (SSMV, red dots) and multi-site (MSMV, green squares) optimised values of calibrated ecophysiological parameters selected based on the multi-objective sensitivity analysis (WS2 in Figure 2). The thick horizontal lines represent medians, boxes represent the interquartile ranges (IQR), and the whiskers represent ±1.5IQR.**

The median reduction of model errors at calibration sites simulated with SSMV parameter sets in comparison to the a priori set was 35 %, 55 %, and 11 % for AbgwC, SoilC, and LitterC, respectively, and 26 %, 35 % and 9 % in comparison to the simulation output obtained with the MSMV parameter set (Figure 7).

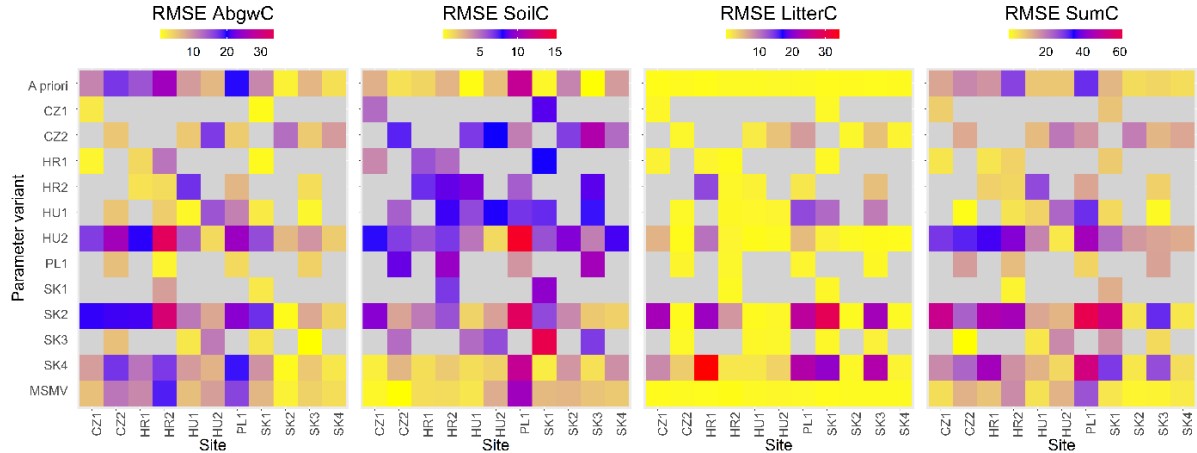

**Figure 7 Heatmaps of root mean square errors (RMSEs) for carbon stocks in aboveground wood, soil, and litter (AbgwC, SoilC, LitterC) and their sum (SumC) for individual calibration sites (X axis) and 13 variants of parameter sets (Y axis): one a priori set, 11 site-specific (SSMV) optimised sets identified by site abbreviations, and one multi-objective (multi-site and multivariate, MSMV) optimised set. The grey colour indicates unsuccessful simulations, which ended with zero or close to zero values of carbon state variables.**

Parameter values obtained from the multi-site optimisation (WS4d in Figure 2) were changed by 2 to 15 % relative to their a priori estimates except for the Sseff, which remained unchanged (Figure 6, Table 2). The MSMV values substantially differed from the SSMV values of most sites, although the differences were parameter specific. The lowest variation was observed for Sseff, while the largest differences between MSMV and SSMV values were found for Nfix (Figure 6). The simulation results obtained using the MSMV parameter set showed reduced mean errors for all three output variables, i.e. AbgwC, SoilC and LitterC, in comparison to the results obtained with the a priori set, by 10, 26, and 5 %, respectively (Table 3, Figure 7).

**Table 3** Evaluation of model performance at 11 calibration sites using the a priori parameter set and the parameter sets optimised with respect to the observed carbon stocks in aboveground wood, soil, and litter (AbgwC, SoilC, and LitterC in kgC m$^{-2}$). "Site-specific" refers to parameter sets derived for single sites using a multivariate approach (SSMV, WS4c phase in Figure 2). "Multi-site" refers to multi-site and multivariate (MSMV) parameter values derived collectively for all calibration sites (WS4d in Figure 2). RMSE = root mean square error, Bias = arithmetic mean of differences between modelled and observed values of the respective variable, MAE = mean absolute error, MPE = mean percentage error, MinDif and MaxDif are minimum and maximum differences between the modelled and observed values of the respective variable.

| Parameter set | Variable | N | RMSE | Bias | MAE | MPE | MinDif | MaxDif |
|---|---|---|---|---|---|---|---|---|
| A priori | | 96 | 9.7727 | 7.3911 | 7.8366 | 29.502 | -8.1569 | 27.007 |
| Site-specific | AbgwC | 96 | 2.0698 | -0.2887 | 1.4873 | 0.012 | -5.4830 | 5.7422 |
| Multi-site | | 96 | 6.8718 | 4.1734 | 5.4798 | 19.246 | -9.5527 | 27.756 |
| A priori | | 72 | 3.5242 | 2.2276 | 2.2630 | 58.329 | -0.2268 | 11.729 |
| Site-specific | SoilC | 72 | 6.8068 | -2.7876 | 6.3782 | -0.408 | -9.7400 | 4.7630 |
| Multi-site | | 72 | 3.0803 | 0.1167 | 2.6423 | 38.068 | -2.4427 | 10.243 |
| A priori | | 10 | 0.2424 | 0.2117 | 0.2117 | 121.774 | 0.0611 | 0.4651 |
| Site-specific | LitterC | 10 | 0.2035 | 0.0030 | 0.1653 | 42.005 | -0.2894 | 0.4382 |
| Multi-site | | 10 | 0.1849 | 0.1417 | 0.1419 | 91.327 | -0.0007 | 0.4185 |

The simulations performed with the site-specific and multi-site optimised parameter sets produced more accurate estimates of carbon stocks in aboveground wood, soil and litter than the a priori set. The non-parametric Wilcoxon signed rank test (Wilcoxon, 1945) with the continuity correction data confirmed insignificant differences between the observed and modelled AbgwC and LitterC simulated with SSMV parameter sets (V = 1807, p = 0.14 for AbgwC, and V = 26, p = 0.92 for LitterC), while the differences for soil carbon were significant (V = 528, p = 1.04e-05, see Sect. 4.2 for explanation). The use of the MSMV parameter set resulted in insignificant differences of SoilC (V = 1396, p = 0.64), while the estimates of AbgwC and LitterC were significantly different from the observations (V = 4094, p = 1.10e-10, and V = 54, p = 0.004, respectively). Nevertheless, the magnitudes of their mean errors calculated for the whole set of calibration sites as well as for most of the individual sites were substantially reduced (Table 3, Figure 7).

### 3.3 Robustness, validation, and plausibility tests

### 3.3.1 Robustness of calibrated parameter values

When simulating the development of calibration sites with the site-specific parameter sets (SSMV) optimised for other sites, we revealed a high variation in modelled outputs per site (Figure 7 and Fig. S5). In 47 % of cases we encountered unsuccessful simulations, during which the modelled forests ceased to exist. Only three SSMV parameter sets led to successful simulations in all calibration sites, but their errors exceeded those obtained with the a priori, respective SSMV or MSMV optimised sets (MPE of AbgwC, SoilC, and LitterC were 39 %, 296 %, and 570 %, respectively).

The robustness test of SSMV and MSMV optimised parameter values using the decision trees (DT) revealed that in 78 % of cases site-specific and muti-site parameter values occurred within the parameters ranges derived from decision trees for individual output variables and sites (Table 4, Fig. S8). The discrepancies between the optimised parameter values and DT ranges occurred for the variables and sites with lower proportions of plausible simulations (Fig. S16) or in the cases when DT

parameter ranges derived for individual output variables did not overlap (e.g. the ranges derived for FLNR and site CR2015, Fig. S8).

**Table 4 Robustness of site-specific (SSMV) and multi-site (MSMV) parameter values based on the analysis of the optimised parameter value occurred within the plausible parameter ranges derived for the individual calibration sites and carbon stocks**
**(aboveground wood, soil and litter carbon labelled as AbgwC, SoilC, LitterC) using decision trees (DT, Sect. 2.5.1). The values for SSMV represent the mean of 11 calibration sites. The value 1 indicates that the MSMV value or all SSMV optimised parameter values occurred within the DT ranges. CLEC = canopy light extinction coefficient, FLNR = fraction of leaf nitrogen in rubisco, MSC = maximum stomatal conductance, Nfix = nitrogen fixation, VPDC = vapour pressure deficit for complete conductance reduction, Sseff = effect of soil stress factor on photosynthesis.**

| | | Proportion of optimised parameter values inside parameter ranges derived using decision tree method | | | | | |
|---|---|---|---|---|---|---|---|
| Output variable | | AbgwC | | SoilC | | LitterC | |
| Parameter set | | SSMV | MSMV | SSMV | MSMV | SSMV | MSMV |
| Parameter | CLEC | 1.00 | 1 | 0.91 | 1 | 0.91 | 1 |
| | FLNR | 0.73 | 1 | 0.64 | 0 | 0.64 | 0 |
| | MSC | 1.00 | 1 | 0.55 | 1 | 0.36 | 0 |
| | Nfix | 0.91 | 0 | 1.00 | 0 | 0.36 | 1 |
| | VPDC | 1.00 | 1 | 1.00 | 1 | 1.00 | 1 |
| | Sseff | 1.00 | 1 | 1.00 | 1 | 1.00 | 1 |
| Mean of All | | 0.94 | 0.83 | 0.85 | 0.67 | 0.71 | 0.67 |

The analysis of interdependencies between the site-specific optimised parameter values revealed the only significant Spearman correlation between CLEC and MSC, which were negatively correlated at 95 % significance level (r=-0.6, p=0.04, Figure 8a). The highest, although non-significant, correlation was found between Nfix and FLNR (r=0.7, p=0.37) suggesting that if Nfix increases FLNR should also increase.

Spearman correlations between parameters and site characteristics revealed that the site-specific optimised values of two calibrated parameters (CLEC, VPDC) were significantly related to elevation (r=0.5 and -0.7, p=0.04, respectively). Significant relationships were also found between CLEC and several climatic variables (Figure 8). The highest positive correlation (r=0.9, p=0.01) of CLEC was with the long-term mean annual precipitation total (AMPRCP), and the highest negative correlation (r=-0.8, p=0.01) with the long-term mean annual vapour pressure deficit (AMVPD). The increasing AMVPD was positively

significantly related to the values of MSC (r=0.7, p=0.05). For other parameters we did not reveal any significant relationships with climate conditions, nor could we confirm any significant correlations of parameter values to soil conditions (Fig. S9),

although positive or negative trends with several environmental characteristics were identified for some parameters, mainly for MSC and VPDC (Figure 8b, Fig. S9).

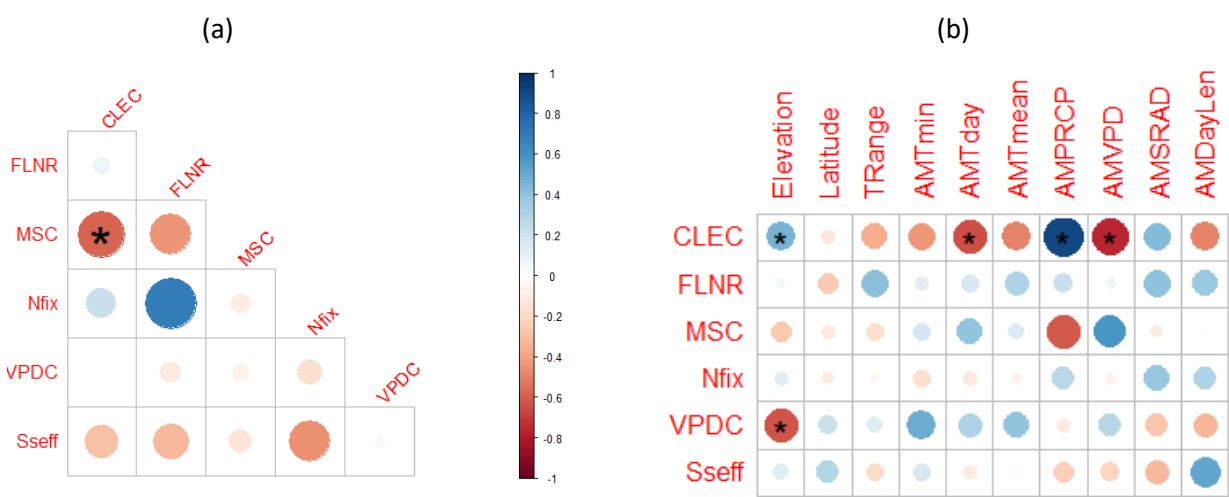

**Figure 8 Spearman correlations between site-specific optimised values of calibrated parameters (a) and between site-specific optimised parameter values and site characteristics (b). The colour and the size of the circles indicate the value of the correlation coefficient. The stars indicate the significance of the correlation (1 star for 95 % significance level, 2 stars for 99 % significance level). CLEC = canopy light extinction coefficient, FLNR = fraction of leaf nitrogen in Rubisco, MSC = maximum stomatal conductance, Nfix = nitrogen fixation, VPDC = vapour pressure deficit for complete conductance reduction, and Sseff = effect of soil stress factor on photosynthesis. Climate characteristics represent long-term annual averages: TRange = Temperature range, AMTmin = Minimum temperature, AMTday = Daylight temperature, AMTmean = Mean temperature, AMPRPCP= Precipitation total, AMVPD = Vapour pressure deficit, AMSRAD = Daily solar radiation, AMDayLen = Daylength.**

### 3.3.2 Model performance with parameter values optimised for multiple sites

The mean absolute error (MAE) between the simulated and observed aboveground wood carbon of 8 validation sites was 0.26 kgC m$^{-2}$ with a 95 % confidence interval from -0.025 to 0.56 kgC m$^{-2}$, while the individual absolute differences varied between -2.06 and 5.11 kgC m$^{-2}$ (Figure 9). The root mean square error was 1.22 kgC m$^{-2}$. The non-parametric Wilcoxon signed rank test with continuity correction indicated non-significant differences between simulations and observations of aboveground wood carbon (V = 1385, p-value = 0.1962). The mean percentage error of AbgwC was 1.25 % of the observed carbon stock in aboveground wood. Hence, both absolute and relative differences were of negligible magnitudes.

Since no observed data on soil and litter carbon were available for validation sites, the simulated SoilC and LitterC were tested against their ranges reported in the literature. The results showed that both variables occurred within the plausible ranges (Fig. S12).

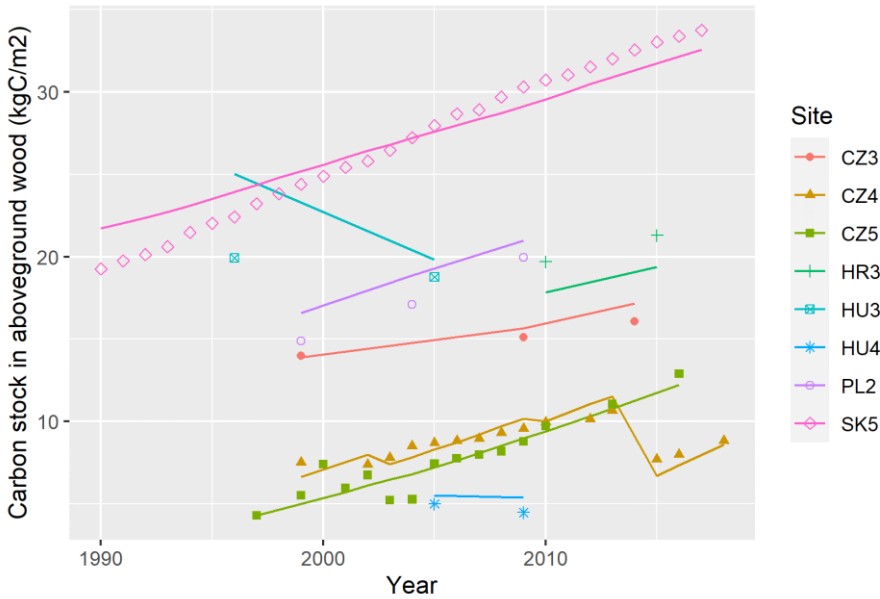

**Figure 9 Comparison of the temporal development of observed (points) carbon stock in aboveground wood (AbgwC) and simulated AbgwC (lines) using the multi-site multivariate optimised (MSMV) parameter set at 8 validation research sites.**

### 3.3.3 Robustness and plausibility of simulated output at a large scale

The simulations of all 87 research sites using the multi-site optimised (MSMV) parameter values were successful and the simulated values of the three output variables (i.e. AbgwC, SoilC, and LitterC) at the end of the spin-up run were well-aligned with the plausible ranges indicated in the literature (Fig. S13).

The simulated values of output variables varied across the studied geographical space (Figure 10) and were significantly correlated with several site characteristics (Figure 11, Fig. S17, Fig. S18). The modelled AbgwC exhibited distinct unimodal responses along the gradients of elevation, long-term mean air temperature, and VPD. The results manifested a production optimum of beech in Central Europe at elevations of 500 - 600 m a.s.l., mean annual air temperature of 9°C, and VPD of 530 Pa. The SoilC and LitterC demonstrated an increasing trend along the elevation gradient and decreasing trends along the climatic gradients. The responses of carbon stocks to soil properties followed a linear pattern, while SoilC and LitterC significantly decreased with the increasing clay content and were positively correlated with the sand content in soil. Aboveground wood carbon was found to be significantly correlated with the proportion of clay in the first and second soil layers, and with the proportion of silt in the fifth layer (Fig. S18). The highest levels of simulated AbgwC were observed on loamy or sandy-loamy soils (Fig. S20).

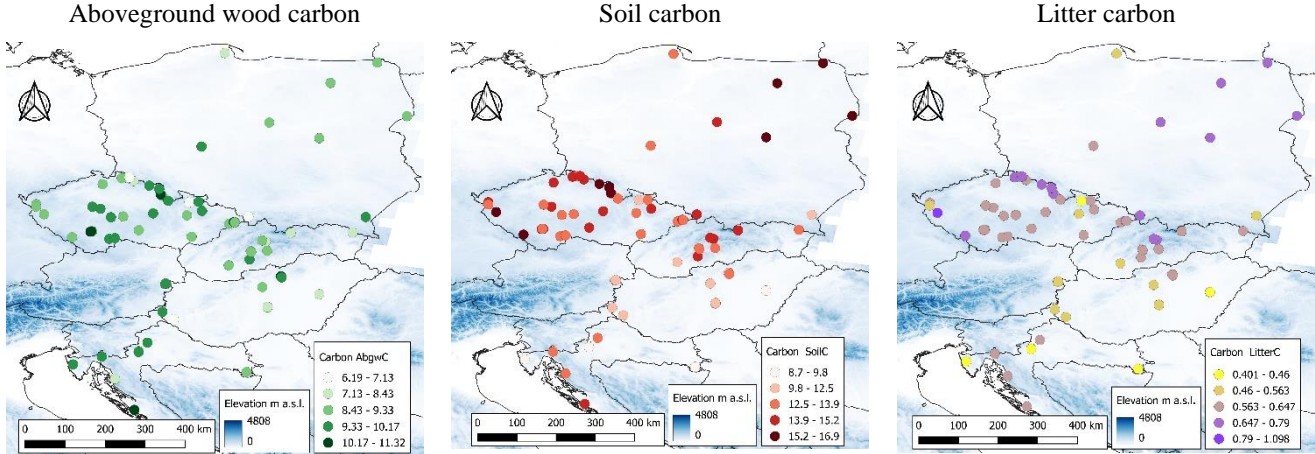

**Figure 10 Simulated values of carbon stocks in aboveground wood (AbgwC), soil (SoilC) and litter (LitterC, all in kgC m⁻²) in beech ecosystems at the standardised stand age of 35 years across 87 sites. The blue background indicates the elevation gradient taken from Hengl et al. (2020).**

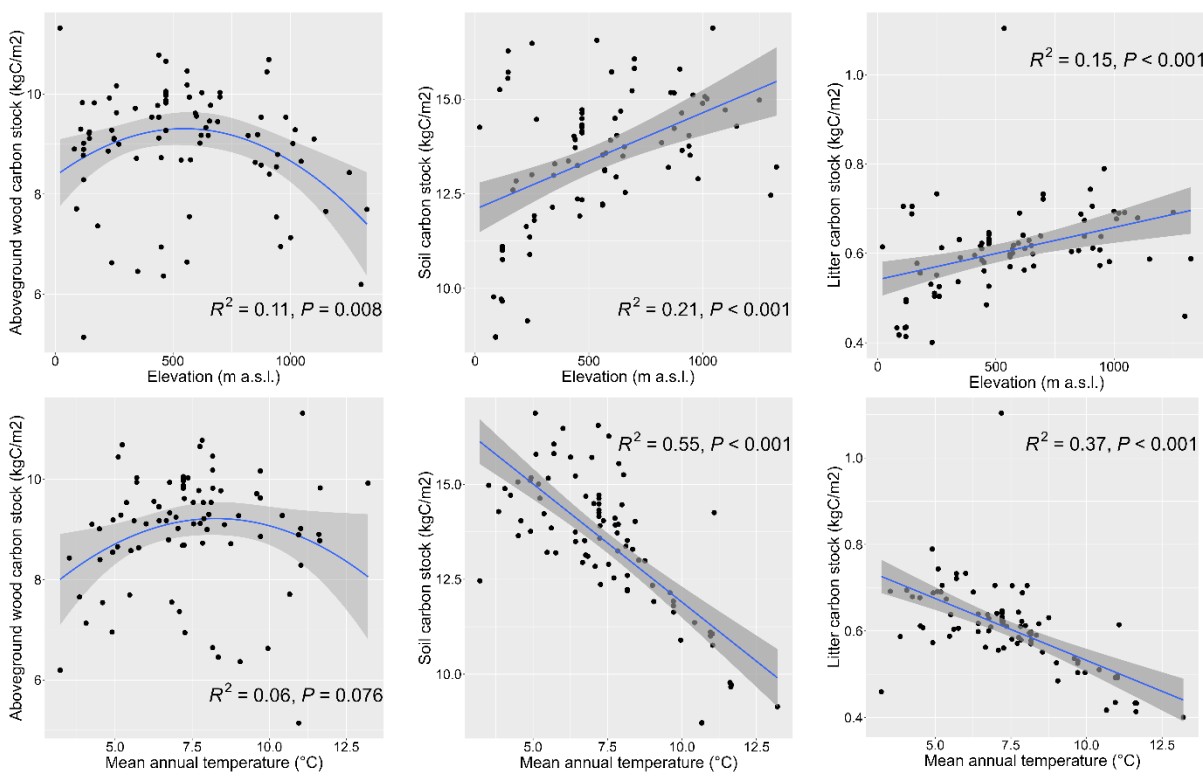

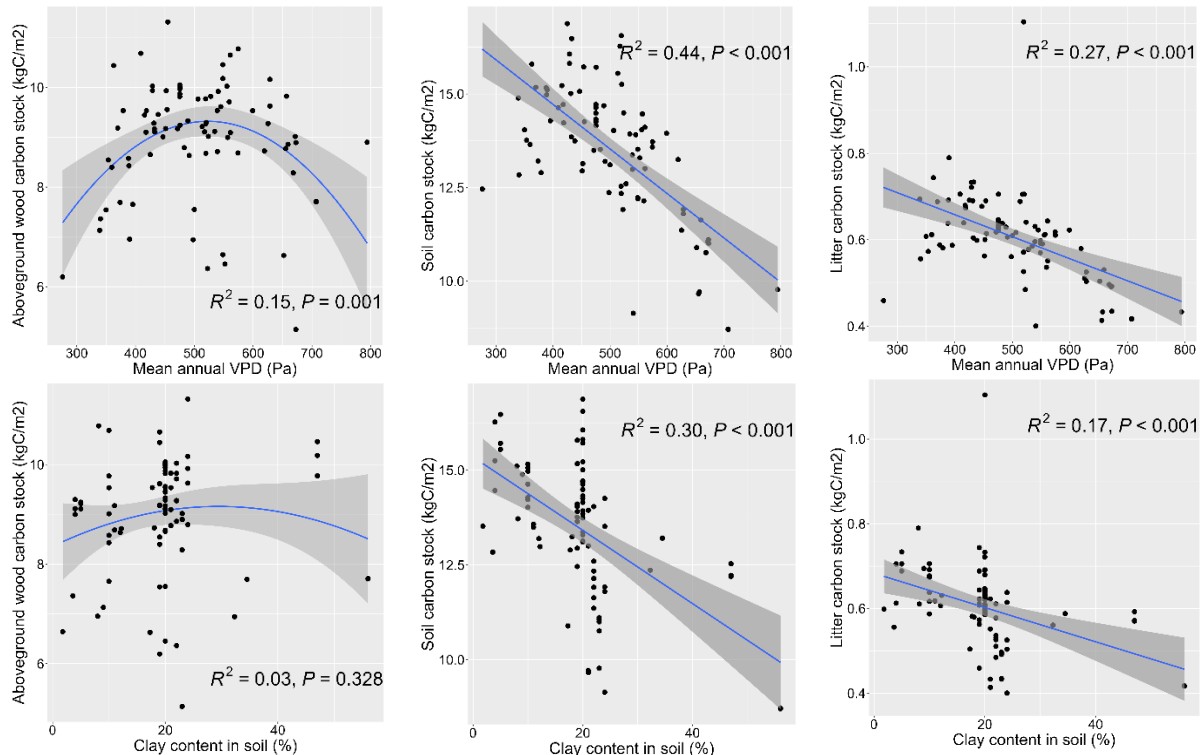

**Figure 11 Responses of modelled carbon stocks in aboveground wood at the standardised stand age of 35 years (left), soil (middle) and litter (right) carbon stocks to selected environmental variables. The simulations were conducted for 87 sites distributed across central Europe (Figure 1). Linear or quadratic regressions were fitted to the data. $R^2$ represents the squared regression coefficient, P is the p-value for the F-test of the fitted model.**

## 4. Discussion

### 4. 1 Selection of parameters for calibration

The global SA showed small differences in the parameter impact between the three examined output variables, because they are all a part of the carbon cycle, which makes them naturally interconnected. The similarity in parameters affecting these output variables underscores the importance of considering a forest holistically, since due to the interactions between different components of the ecosystem changes in one part have cascading effects throughout the system. Therefore, a comprehensive approach that accounts for these interdependencies is crucial for accurate modelling and understanding of forest carbon dynamics.

Several highly influential parameters identified by SA were excluded from the calibration for different reasons. Growth respiration per unit of carbon allocation (GRC), and maintenance respiration per kg of tissue nitrogen (MRperN) were not calibrated because we could not support the modification of their values used in previous model versions (Thornton et al., 2005) by observations. Such data would be needed since the empirical evidence suggests that respiratory parameters may

substantially differ between sites (Lavigne and Ryan, 1997). Other parameters were not included in calibration because the data from literature supported their current values and/or because of the adverse impact on the variables of interest. For example, C:N ratio in leaves was found to have a substantial effect on all examined carbon pools (Figure 5), but to obtain a desired reduction in AbgwC, we would need to increase this parameter from its a priori value (Fig. S2). However, the analysis of the values obtained from site-specific measurements performed at some sites and the data from the TRY database (Kattge et al., 2020) suggested that increasing the value of C:N ratio in leaves would cause a significant deviation from the mean or median of experimental observations (Fig. S3). Similarly, the parameter representing the natural whole plant mortality was not calibrated since we applied dynamic mortality rates during the normal run simulations instead of the temporarily constant mortality. The rates depended on the applied management and were derived from published field observations (Barna et al., 2011; Hülsmann et al., 2018; Pajtík et al., 2018; Vanoni et al., 2019). The possibility of using dynamic mortality rates in BBGCMuSo is a major improvement over the original Biome-BGC model, as mortality has been found to be a driving process of vegetation dynamics in forest growth models (Bugmann et al., 2019; Hlásny et al., 2014).

The identification of influential parameters is crucial not only for model calibration, but also for future studies as it provides valuable information about the key parameters, the values of which should be collected in the field, since applying site-specific values obtained from experimental data may substantially reduce the uncertainty of model simulations. As Thornton et al. (2002) already presented, some parameters, such as C:N ratio in leaves, should be treated as site-specific. In the case the site-specific values of parameters are available, they need to be set prior to the calibration due to the covariance of parameters and need to be excluded from the sensitivity analysis. In our case we aimed at a generic parameter set applicable across the whole studied region, and thus, we used a generalised value of this parameter.

## 4.2 Parameter estimation

The mean error metrics (root mean square errors, mean absolute and percentage errors) showed the increased accuracy of modelled output obtained using single-site (SSMV) and multi-site (MSMV) optimised parameter sets in comparison with those obtained with the a priori parameter set (Table 3). Greater errors of soil carbon for SSMV parameter sets resulted from site-specific very tight positive relationships between the simulated AbgwC and SoilC revealed when analysing Monte Carlo simulations. Due to this, even the high number of performed simulations (100,000 simulations per site) generally covered only a small portion of the 2D space defined by AbgwC and SoilC (Fig. S6). Hence, in some cases improving the results for one of these two carbon pools caused the increase of the error of the other pool within the tested space of six calibrated parameters (Fig. S7). Although some empirical studies reported the positive correlation between aboveground wood and soil carbon stock in the top soil, the relationship is not strong ($R^2$=0.24, Woollen et al., 2012), and frequently insignificant (Osei et al., 2022), as SoilC primarily depends on climate, topography, soil mineralogy and soil texture, especially the content of clay (Powers and Schlesinger, 2002) or sand (Devi, 2021). Our results indicated that different or additional parameters may need to be included

in the calibration that may increase the variability of model output and thus loosen the current high correlations between AbgwC and SoilC.

The application of site-specific calibrated parameter sets outside the respective sites pointed out at the contradiction between their generality and the local accuracy of model estimations. SSMV optimised parameters were not generally applicable, as 47 % of simulations of calibration sites with SSMV parameter sets optimised for different sites collapsed (Figure 7). Calibrating models for individual sites may often result in model overfitting due to the small amount of available data (Tsai et al., 2021), which may lead to completely different parameter sets in the case of a recalibration, and hence a high variance in calibrated 610 values, reducing thus the reusability of calibrated parameter values.

The parameter values optimised for single and multiple sites frequently substantially differed (Figure 6), which indicates the existence of the calibration equifinality, i.e. that many different parameter sets may produce similar output predictions (Beven, 2019). This issue was not apparent at the multi-site level, but occurred at the level of individual sites, at which it can be partially solved using the conditional interval reduction method (CIRM, Hollós et al., 2022). The CIRM approach is based on the 615 iterative narrowing of plausible intervals of parameters using the constraints on the model output. It is an efficient way of dealing with the equifinality unless the contradictions between different output variables occur.

As expected based on similar calibration works of forest growth models that used comprehensive data sources (Forrester et al., 2021; Minunno et al., 2019), the multisite and multivariate calibration increased the generality and robustness of the model application by finding a parameter set that worked across all calibration sites (Figure 7), and the validation set (Figure 9). 620 Nevertheless, differences in data quality and availability across space can substantially influence the calibration results. This problem can be mitigated with calibration techniques that utilise a more appropriate likelihood function (i.e. formal likelihood; Hollós et al., 2022). In this way, the issue of the significant spatial autocorrelation can be resolved using the inclusion of an appropriate error-covariance matrix in the construction of the likelihood function (Tarantola, 2005).

In addition, the multi-site calibration can also reduce spatial heterogeneity of model outputs causing that the average model 625 performance will be good, but at specific sites it can be completely wrong. This could be overcome by applying a hybrid approach that combines the site-specific and multisite calibration. With such an approach, site-specific values obtained from local measurements, well-established relationships derived from large databases or calibrated for specific sites will be used for the parameters that are known to vary in space, while generic values will be used for the other parameters. Thus, the overall correctness can be ensured by multisite calibration, while spatial heterogeneity by site-specific calibration.

Another aspect affecting the calibration is data availability. Although we tried to select calibration sites to cover the whole geographical and environmental space, the central part of the region was overrepresented, while northern parts were not covered due to the insufficient data required for calibration. Nevertheless, the ranges of environmental conditions (Table 1, Figure 1) covered by our data included also extreme sites and seem to represent the natural distribution of European beech (Pagan, 1996) well. To ensure more robust calibration results, more balanced geographical coverage, more long-term data of

multiple variables of interests at individual sites, and a combination of the information about stocks and fluxes at same sites would be required.

## 4.3 Trends in parameters

### 4.3.1 Covariance between parameters

The covariance analysis between parameters found correlations of different magnitudes indicating that in most cases the
parameters were not independent. The revealed significant negative linear relationship between the site-specific values of the canopy light extinction coefficient (CLEC) and the maximum stomatal conductance (MSC, Figure 8) suggested that low values of CLEC should be coupled with high values of MSC and vice versa. However, we have not found any empirical evidence in the literature to confirm or refute the revealed relationship between CLEC and MSC, and therefore it is not clear if the revealed pattern is biologically realistic, or if it is only a side effect of the calibration procedure. Due to this, we did not incorporate this
relationship into the calibration procedure. Another strong (R=0.7) though a non-significant relationship was revealed between nitrogen fixation (Nfix) and fraction of leaf nitrogen in Rubisco (FLNR, Figure 8). Examining this relationship in more detail revealed a non-linear pattern between the two parameters resembling a parabolic curve reaching a maximum of FLNR in the middle of Nfix range (Fig. S10). Similarly as for the previous relationship, we have not found any empirical research dealing with the presented issue although the study by Tang et al. (2019) analysing different species in subtropical ecosystems suggests
that nitrogen fixing trees allocate lower fractions of N to Rubisco than non-nitrogen fixing trees. The FLNR values of different *Eucalyptus* species published by Warren and Adams (2004) do not show any significant trend with the increasing nitrogen amount. The global study by Luo (2021) showed that FLNR is considerably affected by climate and soil factors, including light, atmospheric dryness, soil pH, and sand content. Based on these results, nitrogen fixation does not seem to be directly related to FLNR. Nevertheless, the environmental conditions that affect nitrogen availability can indirectly influence how
nitrogen is allocated within the leaf, including Rubisco, suggesting a complex relationship between them. Still, the pattern of the relationship between FLNR and Nfix across one tree species in temperate ecosystems remains unclear.

Our results indicate the necessity of analysing the covariance between parameters during a model calibration as it not only enlightens the model behaviour and interdependencies between specific parameters but can also increase the efficiency of the calibration procedure by excluding one of the correlated parameters from the calibrated parameter set and estimating its value
only subsequently. In addition, such information may also help to identify the gaps in the available empirical evidence and the direction of future empirical research.

### 4.3.2 Parameter correlations with site characteristics

The analysis of the variation of the optimised site-specific values of parameters across environmental gradients revealed some significant trends (see Figure 8, Fig. S9). This may indicate that a specific parameter should not be kept as a constant but rather as a characteristic that changes depending on driving conditions. An example of such a parameter is the canopy light extinction coefficient (CLEC) that specifies the proportion of solar radiation intercepted in the canopy. A number of process-based models that use this parameter set its value around 0.5, while some models differentiate values between different species (Zhang et al.,

2014). Based on our multi-objective optimisation for beech ecosystems we set the MSMV value of CLEC to 0.66, which is by 10 % higher than the value used by Pietsch et al. (2005) in the original Biome-BGC model for beech (0.6), but it is within the range for broadleaved forests reported by Zhang et al. (2014).

Although most models keep this parameter constant across sites and throughout their simulations (Liu et al., 2021; Zhang et al., 2014), in reality its value changes during a day as well as during a year as it depends on the solar zenith angle, leaf area,

leaf inclined angle, and leaf clumping (Parker, 2020; Wang et al., 2004; Zhang et al., 2014). It also changes with stand age, while it reaches its maximum in young stands (Brown and Parker, 1994). A constant value of CLEC causes intra-annual errors in estimations of plant transpiration and soil evaporation during a year (Tahiri et al., 2006). Due to this, a variable CLEC seems to be a more appropriate option. Our analysis revealed significant trends in the SSMV optimised values of CLEC with multiple environmental characteristics, while the trends with elevation and precipitation were positive, and with temperature and VPD

negative (Figure 8). Such patterns were not observed in other studies analysing measured data of CLEC, but the number of tested observations was low (Zhang et al., 2014) as it was also in our case. Nevertheless, the empirical data showed that CLEC increases with decreasing plant density (Timlin et al., 2014) and LAI (Zhang et al., 2014). Since our model operates at a stand level, stand biomass can be considered as a proxy of stand density. Our results showed the decreasing trend of CLEC with the increasing aboveground wood carbon stock (Fig. S11), which is in agreement with Timlin et al. (2014). Moreover, the

simulated AbgwC of calibration sites decreased along the elevational gradient (Fig. S11) explaining the positive correlation of CLEC to elevation, which can be considered as a side effect of stand density that is lower at high elevations.

Tahiri et al. (2006) successfully applied a simple empirical approach using a linear regression with leaf area. Parker (2020) calculated CLEC as a ratio between the effective LAI and the total LAI. CLEC is usually calculated following the simplified Beer Lambert law as a function of above- and below-canopy solar radiation and leaf area (Zhang et al., 2014). A more

sophisticated approach includes the solar zenith angle and the clumping index. Some models also use the inclination angle of leaves, while most commonly the spherical distribution of leaves is assumed (Liu et al., 2021), although Pisek et al. (2013) found that tree species in temperate and boreal regions are usually characterised by planophile or plagiophile leaf angle distribution.

Although we derived several multiple linear regressions explaining the variation in site-specific CLEC values using different

combinations of environmental predictors (Fig. S15), the simulations with varying CLEC based on environmental conditions did not produce satisfactory results (33 % and 27 % of all and calibration sites collapsed if simulated with CLEC derived from

its regression to annual precipitation, respectively). The possible reason is the existence of interdependencies between parameters discussed in Sect. 4.3.1.

Moreover, we found that MSC was also significantly related to environmental conditions, namely VPD (Figure 8). MSC specifies the highest possible rate at which stomata can open and allow the exchange of gases between the plant and the environment under present-day $CO_2$ concentration and optimal environmental conditions, i.e. maximum radiation, and unlimited water availability, when VPD is zero, and there is no soil water stress. Such conditions rarely occur in the field, and hence, the observed maximum conductance, which represents the highest conductance on fully expanded leaves that was measured during the summer growing season (Murray et al., 2019), does not usually reach the theoretical maximum (McElwain et al., 2016). The theoretical MSC can be derived from leaf anatomy, namely stomatal density, maximum stomatal pore area and stomatal pore depth (McElwain et al., 2016; Murray et al., 2020). The SSMV optimised values of MSC were found to be positively related to the long-term mean VPD (Figure 8), which decreases with elevation, whereas the stomatal conductance as well as stomatal characteristics specifying MSC usually increase with elevation (Bresson et al., 2011; Petrik et al., 2022). In line with this, the studies from temperate European ecosystems reported an inverse relationship between the stomatal conductance and VPD (Körner, 1995; Urban et al., 2017). Similarly, the site-specific values of another ecophysiological parameter representing the stomata closure (VPDC, i.e. the vapour pressure deficit causing the complete conductance reduction) were found to be significantly negatively related to elevation (Figure 8), while the empirical studies did not reveal any differences in the onset of stomatal closure along an elevational gradient (Körner and Cochrane, 1985). Hence, we assume that the revealed correlations in our data are by-products of the site-specific calibrations.

Due to the above above-stated inconsistencies, and the lack of data and supporting information, we decided not to apply the dynamically changing CLEC, MSC and/or VPDC along environmental gradients. However, this approach may be considered as a potential way forward in a future model development when more scientific knowledge becomes available. Nonetheless, our results pointed out that for simulations at a local level, some parameters may need site-specific values. Such a hybrid approach of using a combination of general and site-specific parameters, which was already applied by e.g. Thornton et al. (2002), may be beneficial to reduce the uncertainty of local predictions. Since the values of many of the parameters are usually not measured at research plots, global trait databases, such as TRY (Kattge et al., 2020) or the ones by Liu et al. (2023); Maire et al. (2015); Lin et al. (2015), might be useful to estimate the local values for a specific site and species considering site-specific environmental conditions. Naturally, the best solution for any local study is to obtain measurements of required parameters from specific sites, which is however not always feasible due to time and financial restrictions.

## 4.4 Robustness and plausibility tests of simulated outputs at a large scale

### 4.4.1 Carbon stock in soil

Soil carbon represents a large storage of terrestrial carbon (Amundson, 2001) accounting for approximately a half of total forest ecosystem carbon (Domke et al., 2017; Jobbágy and Jackson, 2000). A similar proportion was also revealed in the output of our simulations (median = 47.7 %, mean = 51.5 %, 1st Qu.= 32.6%, 3rd Qu.= 69.9 %). The absolute values of simulated carbon stock in soil per unit area occurred within the range of soil organic carbon (SOC) reported by empirical studies (De Vos and Cools, 2011; 2010; Wellbrock et al., 2016; Wellbrock and Bolte, 2019), although the variability of simulated values was lower (Fig. S13). The mean value of the simulated SoilC (Min.= 7.9,  1st Qu.=12.1,  median = 13.3, mean = 13.1, 3rd Qu. = 14.4, Max. =17.0 kgC m$^{-2}$) was similar to the mean values observed in European beech forest stands (e.g. Meier and Leuschner, 2010; Mund, 2004).

The site-specific simulated SoilC significantly decreased with the increasing air temperature (Figure 11), which is consistent with the observed patterns in soil carbon stocks from soil profile data along temperature gradients (Hartley et al., 2021; Jobbágy and Jackson, 2000; Post et al., 1982; Sun et al., 2019; Wang et al., 2013). The impact of temperature was also apparent in the relationships between the simulated SoilC and elevation or latitude, both of which were significantly positive (Fig. S17). The increasing trend of SoilC with latitude matched the trend of soil carbon stock found in temperate regions of the northern hemisphere (Minasny et al., 2014). These trends result from the faster decomposition (Wang et al., 2013) and hence the microbial soil respiration as the temperature increases (Cao et al., 2019; Rodeghiero and Cescatti, 2005; Sun et al., 2019), the pattern that was found significant (r=0.33, t = 3.15, p-value = 0.002, 95 % CI = 0.12, 0.51) also in our simulated output (Fig. S19a).

Unlike the increasing trend of SOC with the increasing precipitation reported in the literature (Jobbágy and Jackson, 2000; Post et al., 1982), the simulated SoilC was not significantly related to the precipitation amount (Fig. S19b). In general, the impact of precipitation on SOC changes depending on whether the examined ecosystems are water-limited (Wiesmeier et al., 2019). The small-scale study of SOC in beech forests in Germany revealed its significant correlation to precipitation (Meier and Leuschner, 2010), while at high-latitude ecosystems precipitation has only a minor impact on SOC stock (Devos et al., 2022). Our study includes a much wider variety of environmental conditions including different temperature ranges, soil depths, and soil textures than the study of Meier and Leuschner (2010), which may mask the relationship between SoilC and precipitation. Unfortunately, we could not derive the regional relationships between measured SOC and environmental characteristics from our dataset due to the lack of data on soil carbon stock at all plots. Hence, we performed only plausibility tests with modelled values and compared the revealed trends with those reported in published papers from elsewhere.

When we checked the relationship of simulated SoilC to soil characteristics, we found the opposite trend of SoilC with the increasing clay content (Figure 11) to the one reported in the literature based on soil measurements (Hartley et al., 2021; Jobbágy and Jackson, 2000). The fine mineral fraction composed of medium to fine silt particles and clay is known to have a stabilisation effect on SOC (Hartley et al., 2021), due to which it is often used as an indicator for SOC storage (Wiesmeier et

al., 2019). However, our model results showed that SoilC decreased with the increasing content of fine particles (clay or silt) and increased as sand fraction dominated (Figure 11). Under real conditions, higher clay content supports the formation of soil aggregates that can save organic matter from decomposers and sequester SOC (Angst et al., 2018; Schmidt et al., 2011). In BBGCMuSo this mechanism is not accounted for, as SOC formation is driven solely by temperature and SWC, and the litter input (Hidy et al., 2022). Moreover, the data used for our model simulations did not include the full range of clay content, since the maximum in our database was 56 %, and most site-specific values did not exceed 30 % (median=20 %, mean=19.6 %, 3rd quartile = 22 %). When we experimentally increased the clay content in soils of some sites to the maximum value (i.e. clay content =100 %), we could see the reversed pattern in the relationship (Fig. S19c).

Soil acidity enhances the storage of SOC by reducing soil microbial activities driving the decomposition of soil organic matter (Funakawa et al., 2014). The new BBGCMuSo model includes soil pH as a factor affecting the process of nitrification in soil layers (Hidy et al., 2022). The observed decreasing trend in the simulated output of SoilC with the increasing pH (Fig. S19d) is consistent with the experimental results (Funakawa et al., 2014) confirming the correct implementation of pH impact on soil processes in the model.

Soil carbon is the result of carbon inputs from vegetation followed by decomposition processes. In the model, decomposition is driven by soil temperature and soil water content, which is dependent on the precipitation amount, and water infiltration driven by soil texture (Hidy et al., 2022). In nature, around 60 % of the variability in soil respiration is explained by soil temperature and precipitation (Čater and Ogrinc, 2011). Our simulated output also showed a significantly increasing trend of heterotrophic soil respiration with the increasing temperature (Fig. S19g), but the trends with the soil water content or precipitation were insignificant (Fig. S19h,i). Nevertheless, the increase of the simulated soil water content with the increasing fraction of clay and the decreasing fraction of sand in soil (Fig. S20) was consistent with the general knowledge about the impact of soil texture on soil moisture (Kaufmann and Cleveland, 2008). However, the simulated soil microbial respiration was found to have an increasing though insignificant trend with the increasing clay proportion (Fig. S19e), and a significant decreasing trend with the sand proportion (Fig. S19f). Although these results explain the negative correlation between SoilC and clay content, they contradict our expectations based on the evidence from empirical studies that suggest that decomposition should be faster in coarse-sized soils (Hartley et al., 2021). In addition, soil respiration is strongly driven by root biomass (Čater and Ogrinc, 2011), which was also detected in our simulations (Fig. S19j, significant Pearson's product-moment correlation between the carbon stock in fine roots and heterotrophic respiration with r=0.53, t = 5.74, p-value = 1.45e-07, 95 % CI = (0.36, 0.67)). These findings suggest that while the impact of temperature and vegetation on decomposition is captured well in the model, the influence of soil water seems to be insufficient. Without a thorough data-based analysis it is however not possible to state if the reason lies in the missing process description in the model or in the values of decomposition-related parameters. Nevertheless, the last methodological paper presenting Biome-BGCMuSo (Hidy et al., 2022) also identified decomposition as a process requiring further development.

Similarly to the reported positive relationship of soil carbon to organic carbon input (Cao et al., 2019; Jobbágy and Jackson, 2000), our outputs showed that SoilC increased with the increasing vegetation carbon stock (Fig. S19k), although the correlation was not significant (Pearson's product-moment correlation r=0.15, t = 1.43, p-value = 0.16, 95 % CI= (-0.06, 0.36)). In the model direct carbon inputs into soil storage come from the litter (Hidy et al., 2022). The significant positive correlation

(r=0.89, t=16.79, p-value < 2.2e-16, 95 % CI= (0.82, 0.92)) between the simulated litter and soil carbon stocks (Fig. S19l), confirmed that the model captured the carbon flow from vegetation to the soil according to expectations based on published field data (Hilli et al., 2010). In the model, litter is formed by leaf fall, fine root mortality, and defragmentation of coarse woody debris (CWD, Hidy et al., 2022). Surprisingly, the relationships of simulated SoilC to the annual amount of carbon in leaves or fine roots were insignificant (Fig. S19m,n), while the correlation of SoilC with CWD was significantly positive (Fig.

S19o). These results were caused by a much higher amount of accumulated CWD in the simulated ecosystem in comparison to the input from leaves or fine roots. Hence CWD represents the main source of carbon for soil. In the current model version, the actual amount of CWD results from the accumulation over the whole simulation that cannot be reduced by a user, although the usual practice in managed forests has been to remove dead trees during logging operations for sanitary reasons (Kirby et al., 1998; Paletto et al., 2012). Hence, the actual amount of CWD found in managed forests normally represents only a small

fraction of the CWD stock of what can be found in nature reserves (Christensen et al., 2005). To make the ecosystem development under human influence more realistic, the future model version will include the possibility to simulate the extraction of CWD or its part from the system at any time during the normal run simulation.

### 4.4.2 Carbon stock in litter

The absolute values of simulated carbon stock in litter (Min. = 0.33, 1st Qu.= 0.53, Median = 0.59, Mean = 0.61, 3rd Qu. =

0.68, Max. = 1.07 kgC m$^{-2}$) were consistent with the litter carbon stock reported from beech forests in Europe (Meier and Leuschner, 2010; Mund, 2004; Vesterdal et al., 2008). The litter amount represented approximately 2.5 % of the total ecosystem carbon (Min.= 0.6, 1st Qu.= 1.4, Median = 2.2, Mean= 2.5, 3rd Qu.= 3.7, Max.= 9.8 %), which is lower than the relative mean litter C stock reported globally (5 % based on Pan et al. (2011)) or for the U.S.A. (7 %, Domke et al., 2016). Such a relatively small amount of organic litter is typical for temperate hardwood forests on fertile soils (BMELF, 1997).

We revealed similar trends of LitterC along environmental gradients as for SoilC, e.g. the simulated carbon stock in litter significantly decreased with the increasing temperature since the heterotrophic respiration also increases along the temperature gradient (Sun et al., 2019). Similar trends were also found with elevation, and latitude, as well as with soil characteristics (Fig. S17 and Fig. S18). The decreasing trends of simulated LitterC with the increasing pH, clay and silt proportion in soil and with the decreasing content of coarse sand were consistent with the trends derived from field measurements (Vesterdal and Raulund-

Rasmussen, 1998). Based on the empirical evidence by Meier and Leuschner (2010), fine root biomass is the major factor affecting carbon stock in litter. However, our analysis did not reveal a significant relationship between carbon stock in fine roots and litter (Fig. S19p) probably due to the differences in the perception of the term "litter" in the model and in empirical studies. Vesterdal and Raulund-Rasmussen (1998) also found significant correlations of LitterC to the soil content of other

chemical elements (P, Ca, K, Mg), which are not included in our model and in most available models of vegetation dynamics
(Merganičová et al., 2019). Nevertheless, due to the ongoing climate change including the dynamics of other nutrients in models may become more important especially if they represent limiting factors for ecosystems (Zaehle, 2013).

### 4.4.3 Carbon stock in aboveground wood biomass

The accumulated carbon stock in wood biomass strongly depends on the forest age or the forest developmental phase. Due to
830 this we first compared the absolute values of AbgwC at the end of spinup simulations to the stock observed in over-aged and old-growth beech forests (Barna et al., 2011). The absolute values occurred within the reported range, although the variability of simulated values was substantially lower than in the observed ones (Fig. S13). On average, around 40 % of total ecosystem carbon was fixed in simulated AbgwC (1st Qu.=21.6, median=43.5, mean=39.9, 3rd Qu.=57.9, Max.=78.7 %), similarly as reported from temperate European forest ecosystems (Wellbrock et al., 2017).

Simulated values of AbgwC exhibited a parabolic relationship to elevation and temperature, with maximum values at elevations of around 500 - 600 m a.s.l., and a mean annual air temperature of approximately 9°C (Figure 11). These results coincide with optimum growth conditions for beech reported in the literature, based on which beech growth optimum occurs between 450 and 900 m a.s.l. (Schieber et al., 2013), and at sites with a mean annual temperature of 7 to 10°C (Czajkowski et al., 2006; Pagan, 1996; Paule, 1995). The literature also suggests the optimum annual precipitation total for beech from 700 to
1,000 mm (Czajkowski et al., 2006; Paule, 1995), but the relationship of the simulated AbgwC with precipitation explained only 5 % of AbgwC variability. Nevertheless, the unimodal relationship of AbgwC with VPD revealed the maximum AbgwC at around 530 Pa of VPD (Figure 11), which falls within the optimum VPD range for plant growth (500 to 1,200 Pa, Noh and Lee, 2022). Although empirical studies reported an inverse relationship of beech production to VPD (Lendzion and Leuschner, 2008; Roibu et al., 2022; Tumajer et al., 2022), they focus on short-term changes, whereas in our analysis we used a long-
term mean VPD characterising overall site conditions. Already Leuschner (2002) showed in his experiment that the prevailing VPD during the plant development determines the growth potential of plants under the conditions of Central Europe. The laboratory experiment by Lihavainen et al. (2016) revealed that the effect of VPD changes in time. While the initial reduction of VPD to low values caused an acceleration in the growth rate of silver birch, the effect diminished in time due to nitrogen limitation (Lihavainen et al., 2016). Since VPD seems to have a more profound effect on an intra-annual growth of broadleaved
tree species than temperature (Tumajer et al., 2022), more research is required to clarify the impacts at different temporal levels.

The simulated AbgwC trend along the soil gradient was consistent with the empirical knowledge about optimum soil conditions for beech. Beech prefers well-drained soils and does not tolerate wet clay soils. It frequently occurs on loamy or sandy-loamy soils (Packham et al., 2012), on which the simulated AbgwC was the highest (Fig. S20). Loams are the most productive soils
because of their moderate soil texture due to which they are able to hold a large amount of water available for plants (Kolb,

2022). Soil texture also affects fine root production, e.g. Weemstra et al. (2017) observed significantly higher fine root biomass on sand than on clay. We did not reveal such a trend in our simulation outputs (Fig. S19q, Pearson's product-moment correlation r=0.11, t = 1.0, df = 82, p-value = 0.32, 95 % CI= (-0.11, 0.32)) because C allocation in the model is fixed and does not depend on soil texture. Although beech forests grow on soils with a large range of pH from 3.5 to 8.5 (Packham et al., 2012), the

optimum values at which the maximum biomass production is achieved fluctuate between 5.5. and 6 (Pagan, 1996). The pH of the plots in our database varied between 3.69 and 7.5 (1st Qu.= 5.1, median = 5.2, mean = 5.6, 3rd Qu. =  6.4), but we did not reveal a significant trend in AbgwC with pH in the simulated output (Fig. S19r).

### 4.5 Future model development

Model structural uncertainty and parameter uncertainty are not distinguishable. Inevitable structural uncertainties exist in

Biome-BGCMuSo and essentially in all other process-based models, which means that the processes are simplified, and some internal processes can compensate each other. We typically call this phenomenon as getting good results for wrong reasons.

The variability in output data along the gradients of individual characteristics indicates the complex nature of the model and the combined impact of multiple environmental and ecosystem conditions on the final state of the system. In general, we can say that the model output behaves according to well-known natural rules along environmental gradients. The revealed

discrepancies are of lesser importance and point out at the issues in the model that should be dealt with in the future model development. Water seems to play a minor role in modifying the simulated carbon-related output, but this requires more thorough tests using data capturing the water cycle that were not used in the current study. Similarly, the impact of soil texture needs to be examined in more detail to drive the conclusion. Moreover, there are environmental characteristics which are not accounted for in the current model but may explain the differences between the observed and modelled trends in soil carbon

stock, e.g. parent rock material (Wiesmeier et al., 2019), or the proportion of coarse rock fragments in soil that may substantially influence soil properties, such as water holding capacity and movement, plant growth and decomposition processes (Poesen and Lavee, 1994). In the current study we have not addressed these issues due to the lack of field observations.

Carbon cycling in simulated forest ecosystems is primarily driven by allocation, respiration, mortality, and decomposition.

The relationships between individual output variables representing the carbon cycle are in general consistent with the empirically based knowledge. Including the possibility of CWD removal from an ecosystem due to management interventions will enable more realistic simulations of managed forests and should also result in better capturing the relationships between SoilC and carbon in foliage or fine roots.

Although in the model, a forest is represented by two leaves, one sunlit and one shaded one, this has implications only on the

calculations of photosynthesis, while the other processes are not separated between the two parts. Due to this, the parameter called the ratio of shaded specific leaf area (SLA) to sunlit SLA did not currently have a substantial effect on the examined carbon stocks, especially in aboveground wood (Table 1). Future model development could account for the differences in

respiration and allocation proportions and mortality of overstorey and understorey. This would enhance the model applicability to simulate the development of two-storeyed forests and should also increase the variability of model output due to the differences in the growth efficiency between the forest storeys.

Another limitation is the fixed C allocation over the whole simulated period driven by species-specific C allocation parameters. This approach is the simplest one (Merganičová et al., 2019) and was found sufficient when simulating short-term dynamics of ecosystems. However, for multi-site simulations covering long-term dynamics, a fixed C allocation may lead to bias in model output at certain sites or during certain developmental phases of forests, which may require site-specific or phase-specific parameter values.

Other structural improvements needed in Biome-BGCMuSo include improved N cycle and consideration of additional SOC decomposition mass flows including root exudates, priming and litter decomposition to avoid the bias in the estimated parameters. It is a major challenge to the community, and it is not foreseen that the parameter estimation will ever be free from errors.

## 5. Conclusion

This work presented a novel multi-objective calibration approach that uses the generalised likelihood uncertainty estimation method, plausibility checks of output variables, and intersection principles. The proposed approach solves the problems of model overfitting, calibration efficiency, spatial heterogeneity, and the amount and quality of available data. The sensitivity results highlighted the need for the multivariate approach, as the impact rates of parameters and the trends of changes differed between the selected output variables. The integration of the plausibility checks of model outputs ensures the realism of the simulated dynamics. The most important advantage of the presented method is that it considers the environmental dependency of ecophysiological parameters in a spatial context. Moreover, the approach is also usable to select site (environment) invariant parameters, that are globally applicable. Another advantage compared to traditional Bayesian or frequentist methods is the plausibility check of the optimised parameters and their ranges, where global databases on plant traits play a crucial role. The solution improves the reliability of the optimisation and may be generally applicable to any process-based model of ecosystem dynamics.

The disadvantage of the optimisation method is the possible bias of optimised model parameters that can occur because the parameters are forced towards specific values during the optimisation process to match the simulated output with observations. This can be partially avoided by including multiple data into calibration, which represent diverse parts of an ecosystem, such as vegetation, litter, and soil, as it was presented in this study, simulated nutrients (in our case it would be carbon, nitrogen, and water), and processes. To identify the bias in parameter values, on-site measurements of parameters would be needed. Hence, it is worth considering obtaining the information on some plant characteristics, e.g. C:N ratios in different ecosystem compartments, FLNR, etc., routinely from research plots.

The presented optimisation method can be further enhanced. In this study the likelihood function did not include the uncertainty of the observations, which means a lack of weighting of errors with respect to their magnitudes. Thus, the likelihood can be reformulated to include the observation uncertainty. Moreover, the method does not currently account for the covariance between output variables. This could be done by constructing a covariance matrix representing the relationships between the output variables from the model simulations and incorporating it into a multivariate likelihood function. Such an approach could provide a more accurate and realistic estimation of the uncertainties associated with model parameters. On the other hand, including covariance would further increase the computational complexity of the method, which is already high.

The calibration of the model performed at individual sites (SSMV) and multiple sites (MSMV) revealed pros and cons of both approaches. Site specific parameter values improved the accuracy of the simulated outputs of interest for the specific sites and thus, are more suitable for local simulation studies than the generalised parameter set, which is more appropriate for studies covering a larger spatial scale.

The independent validation, robustness and plausibility tests confirmed the robustness of the multi-site and multivariate calibrated set of ecophysiological parameters for the European beech at a regional level. The study highlighted the gaps in the empirical data and knowledge explaining the relationships between parameters, or between parameters and environmental conditions, which should be addressed by future research. For future applications, additional parameters that were not considered in this study, such as parameters specifying drought-induced mortality, may need to be calibrated with additional empirical data since the occurrence of extreme events and disturbances has been increasing due to the climate change.

**List of tables**

Table 1 Summary of site and forest stand characteristics for the whole data set (All), calibration and validation sets. Climate data represent a period from 1950 to 2018.

Table 2 List of parameters tested and modified during the calibration of Biome-BGCMuSo for *Fagus sylvatica* L. Columns Min and Max represent the minimum and maximum values used to define the parameter ranges for the sensitivity analysis and optimisation. Local Sensitivity contains the values of the sensitivity index per output variable (carbon stock in aboveground wood, soil, and litter labelled as AbgwC, SoilC, and LitterC) calculated following Hoffman and Gardner (1983, Eq. 1). The column "First" is the first parameter approximation identified based on the species-specific values derived for Biome-BGC by Pietsch et al. (2005) (in italics) and the complementary literature review. The column "A priori" contains parameter values that enabled the successful simulations of all 87 sites(result of the WS1 phase in Fig. 2). The column "MSMV" contains the multi-site and multivariate calibrated parameter set (result of the WS4d phase in Fig. 2) based on 11 calibration sites and 3 output variables (AbgwC, SoilC, and LitterC). The calibrated parameters are indicated with the grey background.

Table 3 Evaluation of model performance at 11 calibration sites using the a priori parameter set and the parameter sets optimised with respect to the observed carbon stocks in aboveground wood, soil, and litter (AbgwC, SoilC, and LitterC in kgC m$^{-2}$). "Site-specific" refers to parameter sets derived for single sites using a multivariate approach (SSMV, WS4c phase in Fig. 2). "Multi-site" refers to multi-site and multivariate (MSMV) parameter values derived collectively for all calibration sites (WS4d in Fig. 2). RMSE = root mean square error, Bias = arithmetic mean of differences between modelled and observed values of the respective variable, MAE = mean absolute error, MPE = mean percentage error, MinDif and MaxDif are minimum and maximum differences between the modelled and observed values of the respective variable.

Table 4 Robustness of site-specific (SSMV) and multi-site (MSMV) parameter values based on the analysis of the optimised parameter value occurred within the plausible parameter ranges derived for the individual calibration sites and carbon stocks (aboveground wood, soil and litter carbon labelled as AbgwC, SoilC, LitterC) using decision trees (DT, Sect. 2.5.1). The values for SSMV represent the mean of 11 calibration sites. The value 1 indicates that the MSMV value or all SSMV optimised parameter values occurred within the DT ranges. CLEC = canopy light extinction coefficient, FLNR = fraction of leaf nitrogen in rubisco, MSC = maximum stomatal conductance, Nfix = nitrogen fixation, VPDC = vapour pressure deficit for complete conductance reduction, Sseff = effect of soil stress factor on photosynthesis.

Figure 1 a) Distribution of used forest sites across Europe with forest cover displayed in the background (CORINE Land Cover, 2023), b) The coverage of the entire climatic space of European beech in Europe (represented by grey dots using the data from Caudullo et al. 2017) by the used forest sites. Black dots indicate 87 sites used for testing model plausibility. Red triangles indicate eleven data-rich sites used for the calibration of the BBGCMuSo model. Blue crosses and diamonds represent eight validation sites.

Figure 2 A general workflow used for the optimisation of ecophysiological parameters of BBGCMuSo for the European beech using the multi-objective calibration approach.

Figure 3 Identification of site-specific feasible parameter ranges using a multivariate approach (a); and identification of multi-site and multivariate optimised (MSMV) parameter values and their parameter ranges (b). The approach in (a) is demonstrated using an example for a single site, a single parameter and two variables. The tested parameter interval is indicated by the grey dotted pattern, feasible parameter ranges for individual variables are indicated with blue and green colours, and the multivariate feasible parameter range if shown in orange. The approach (b) is demonstrated using a hypothetical 2D parameter space and site-specific optimised parameter ranges for two sites. The grey dotted space indicates the tested 2D parameter space, the blue and green colours refer to 2D parameter space for specific sites, and the orange space refers to multi-site and multivariate feasible parameter ranges. Min = minimum, Max = maximum, Opt = optimised values of tested parameters.

Figure 4 Determination of plausible ranges of parameters using decision trees across all sites and output variables. Par = parameter, background transparent coloured horizontal lines represent the tested parameter range, while non-transparent coloured lines show the optimised parameter ranges based on decision trees performed for carbon stock in aboveground wood (AbgwC), soil (SoilC), and litter (LitterC). The dashed red rectangle represents the plausible range of a single parameter based on the intersection of its plausible ranges derived for the three output variables.

Figure 5 Results of global (multi-parameter) sensitivity analysis (WS2b in Fig. 2) of simulated carbon stock in aboveground wood, soil and litter (AbgwC, SoilC, LitterC) to ecophysiological parameters of BBGCMuSo. Parameter abbreviations are given in Table 2. Dark grey rectangles with black arrows above indicate parameters selected for the calibration procedure: canopy light extinction coefficient (CLEC), fraction of leaf nitrogen in rubisco (FLNR), maximum stomatal conductance (MSC), nitrogen fixation (Nfix), effect of soil stress factor on photosynthesis (Sseff), and vapour pressure deficit for complete conductance reduction (VPDC).

Figure 6 A priori (blue triangles), site-specific (SSMV, red dots) and multi-site (MSMV, green squares) optimised values of calibrated ecophysiological parameters selected based on the multi-objective sensitivity analysis (WS2 in Fig. 2). The thick horizontal lines represent medians, boxes represent the interquartile ranges (IQR), and the whiskers represent ±1.5IQR.

Figure 7 Heatmaps of root mean square errors (RMSEs) for carbon stocks in aboveground wood, soil, and litter (AbgwC, SoilC, LitterC) and their sum (SumC) for individual calibration sites (X axis) and 13 variants of parameter sets (Y axis): one a priori set, 11 site-specific (SSMV) optimised sets identified by site abbreviations, and one multi-objective (multi-site and multivariate, MSMV) optimised set. The grey colour indicates unsuccessful simulations, which ended with zero or close to zero values of carbon state variables.

Figure 8 Spearman correlations between site-specific optimised values of calibrated parameters (a) and between site-specific optimised parameter values and site characteristics (b). The colour and the size of the circles indicate the value of the correlation coefficient. The stars indicate the significance of the correlation (1 star for 95 % significance level, 2 stars for 99 % significance level). CLEC = canopy light extinction coefficient, FLNR = fraction of leaf nitrogen in Rubisco, MSC = maximum stomatal conductance, Nfix = nitrogen fixation, VPDC = vapour pressure deficit for complete conductance reduction, and Sseff = effect of soil stress factor on photosynthesis. Climate characteristics represent long-term annual averages: TRange = Temperature range, AMTmin = Minimum temperature, AMTday = Daylight temperature, AMTmean = Mean temperature, AMPRPCP= Precipitation total, AMVPD = Vapour pressure deficit, AMSRAD = Daily solar radiation, AMDayLen = Daylength.

Figure 9 Comparison of the temporal development of observed (points) carbon stock in aboveground wood (AbgwC) and simulated AbgwC (lines) using the multi-site multivariate optimised (MSMV) parameter set at 8 validation research sites.

Figure 10 Simulated values of carbon stocks in aboveground wood, soil and litter (kgC m$^{-2}$) in beech ecosystems at the standardised stand age of 35 years across 87 sites. The blue background indicates the elevation gradient taken from Hengl et al. (2020).

Figure 11 Responses of modelled carbon stocks in aboveground wood at the standardised stand age of 35 years (left), soil (middle) and litter (right) carbon stocks to selected environmental variables. The simulations were conducted for 87
sites distributed across central Europe (Fig. 1). Linear or quadratic regressions were fitted to the data. R$^2$ represents the squared regression coefficient, P is the p-value for the F-test of the fitted model.

**Code and data availability.** The current version of Biome-BGCMuSo, together with sample input files and a detailed user guide, is available from the website of the model at http://nimbus.elte.hu/bbgc/download.html under the GPL-2 license. Biome-BGCMuSo v6 is also available at GitHub: https://github.com/bpbond/Biome-BGC/tree/Biome-BGCMuSo_v6. The exact version of the model (v6.2 alpha) used to produce the results in this paper is archived on Zenodo (https://doi.org/10.5281/zenodo.5761202; Hidy and Barcza, 2021). The RBBGCMuSo package (Hollós et al., 2023) is available at https://github.com/hollorol/RBBGCMuso. Experimental data used in the study are available from ICP Forests (http://icp-forests.net/) and from authors that provided the data upon request. The code for optimisation of parameters is in the supplement.

**Supplement**. The supplement related to this article is available online.

**Author contributions**. KM and TH conceived and designed the study and KM carried the experiments with the assistance from JM, LD, and RH. DH and ZB developed Biome-BGCMuSo and maintained the source code. RH and ZB developed RBBGCMuSo. KM and DK performed the literature review. KM executed simulations, and KM and JM performed the analyses. LD and JM contributed with model benchmarking. ZS, PP, HM, JN, MB, DB, MZOS, and PF contributed with experimental data. KM prepared the paper and the Supplement with contributions from all co-authors. KM, JM, TH, ZB, RH, MZOS, HM, and DB revised the paper based on the reviews of two anonymous reviewers. All authors reviewed and approved the present article and the Supplement.

**Competing interests.** The authors declare that they have no conflict of interest.

**Acknowledgements**

The study was supported by COST PROCLIAS CA19139, grant "EVA4.0", No. CZ.02.1.01/0.0/0.0/16_019/0000803 financed by OP RDEEVA4.0, and the Slovak Research and Development Agency, projects No. APVV-21-0412 (KM, PF), APVV-18-0305, APVV-22-0001 (JM), APVV-19-0183 (MB). HM, MZOS, and DB were supported by the Croatian Science Foundation Project MODFLUX (HRZZ IP-2019-04-6325). ZB, RH and DH were supported by the National Multidisciplinary Laboratory for Climate Change, RRF-2.3.1-21-2022-00014 project. PP was supported by „TreeAdapt" funded on the base of contract between the National Forest Centre and Ministry of Agriculture and Rural Development of the Slovak Republic. We thank two anonymous reviewers for their valuable comments that substantially improved the manuscript.

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

Fig. S11 Linear regressions between site-specific optimised values (SSMV) of canopy light extinction coefficient (CLEC) and carbon stock in the aboveground wood at the standardised age of 63 years (AbgwC) and elevation of 11 calibration sites.

Fig. S12 Validation of multi-sites multivariate optimised parameter set based on the simulations of 8 research sites. The figures represent boxplots of the differences between the modelled and observed aboveground wood carbon stock (AbgwC, left), and modelled carbon stock in soil (SoilC, middle) and litter (LitterC, right) compared to ranges observed in the field (horizontal red lines represent median (solid line) and 5 and 95% percentiles (dashed lines)). The ranges for carbon stock in soil and litter were taken from (Pavlenda and Pajtík, 2010). The thick horizontal lines of boxplots represent medians, boxes represent the interquartile ranges (IQR), and the vertical lines represent ±1.5IQR.

Fig. S13 Plausibility of carbon simulated content in aboveground wood, soil and litter (AbgwC, SoilC, LitterC) at the end of spinup simulations performed for all sites in comparison to ranges derived from the literature (horizontal red lines represent median (solid line) and 5 and 95% percentiles (dashed lines)). The ranges for carbon stock in the aboveground wood were taken from (Barna et al., 2011) for over-aged beech forests at average-quality sites. The ranges for carbon stock in soil and litter were taken from (Pavlenda and Pajtík, 2010).

Fig. S14 Responses of modelled carbon stock in the aboveground wood (AbgwC) at the standardised stand age of 35 years (left), soil (middle) and litter (right) carbon stocks to sand content in soil (top) and soil texture (bottom). The simulations with BBGCMuSo were conducted for 87 sites distributed across central Europe (see Figure 1).

Fig. S15 Multiple linear regressions explaining the variation in site-specific canopy light extinction coefficient (CLEC) values using different combinations of environmental characteristics. The size of the model represents the number of predictors included in the multiple regression model. The figure on the left presents the selected combinations of environmental characteristics, while black squares indicates, which characteristics are included in multiple regression models and black dots indicate their absence in the model. The right figure shows statistical characteristics of derived models as follows: R2—R-squared, R2adj—adjusted R-squared, AIC—AKAIKE information criterion, BIC—Bayesian information criterion, Cp—Mallows' statistic, Sigma—residual standard deviation. The abbreviations of environmental characteristics are explained in Table S1.

Fig. S16 Plausibility percentages of BBGCMuSo simulations per calibration site and carbon stock in aboveground wood, soil, litter and all output variables at the same time (AbgwC, SoilC, LitterC, All).

Fig. S17 Spearman correlations between selected carbon-related output variables and site characteristics. The colour and the size of the circles indicate the value of the correlation coefficient, and the stars indicate the significance of the correlation (1 star, 2 stars and 3 stars represent 95%, 99% and 99.9% significance levels, respectively). The abbreviations are explained in Table S1.

Fig. S18 Spearman correlations between selected carbon-related output variables and soil particle proportions in individual soil layers. The colour and the size of the circles indicate the value of the correlation coefficient, and the stars indicate the significance of the correlation (1 star, 2 stars and 3 stars refer to 95%, 99% and 99.9% significance levels, respectively). The abbreviations are explained in Table S1. Values behind soil particle groups indicate individual soil layers, while Aver refers to the mean value calculated from all layers.

Fig. S19 Relationships between selected output variables and environmental characteristics derived from the BBGCMuSo simulated output of 87 sites distributed across central Europe (see Figure 1).

Fig. S20 Relationships between the simulated volumetric soil water content and soil texture based on the BBGCMuSo simulated output of 87 sites distributed across central Europe (see Figure 1).