# Peer review of "Biogeochemical model Biome-BGCMuSo v6.2 provides plausible and accurate simulations of carbon cycle in Central European beech forests"

_Geoscientific Model Development, 2024_

## Referee Comment (RC1)

Review of Merganičová et al., "Biogeochemical model Biome-BGCMuSo v6.2 provides plausible and accurate simulations of carbon cycle in Central European beech forests"

**General comments:**

This paper describes a method of parameterization and application of the Biome-BGCMuSo v6.2 model for European beech forests and evaluates its accuracy in simulating aboveground wood, soil, and litter carbon stocks. In general, the paper is well-written and offers a useful methodology for setting parameters in biogeochemical models. It is very challenging to model carbon stocks correctly on a continental scale, and the authors have produced reasonable results for a large area in eastern Europe spanning multiple climatic zones. However, there are a few places where the manuscript could benefit from clarification and/or expanded discussion. I would like to see a section that clearly outlines the advantages and disadvantages of the parameterization methodology. It is currently discussed but scattered in various parts of the manuscript. It would also be good to focus the discussion more on general lessons—what processes are most important for accurate simulations, and what's missing or inadequately represented in the current version of the model— and priorities for future model development. There are also some minor corrections needed as specified below.

**Specific comments:**

Abstract

Line 28: "model wide-ranging applicability" sounds awkward; rephrase to something like, "applicability of the model over a wide range".

1. Introduction

The first few paragraphs (lines 52 – 71) contain a lot of generalizations and model development history that are not particularly helpful in introducing the topic of the manuscript. I suggest you cut most of this and focus specifically on the challenges of simulating carbon dynamics in forest ecosystems and different approaches for calibrating parameters.

Line 74: "intrinsic variability" isn't quite right here, you are describing uncertainty and range that comes from mostly external factors that are not intrinsic to the system

Lines 77-79: awkward sentence, rephrase

Lines 86-87: replace "generality" with "wide applicability"; elaborate on the "advanced calibration designs"—what needs improvement, and what advantages does your approach offer?

Line 89: replace "addressed" with "used"

Line 91: "these elements"—specific what these are (e.g. carbon, nitrogen, etc)

Line 97: delete "the" at end of line

Line 105: not sure what is meant by "phenotypical plasticity", please explain/elaborate

2. Data and Methods

2.1 Model: This section would benefit from the addition of more details about the model, such as the timestep (daily) and the main inputs (meteorology and soil information). While the next subsection 2.2 mentions the datasets used, it's not clear from the text that these are required to drive the model.

Lines 116-117: It's not clear what is meant by a "two-leaf" representation

Line 128: "storages" should be singular, i.e. "storage"

Lines 130-132: Here you are implying that there are many crop-specific parameters that were added, but are these needed to simulate forests?

Lines 150- 157 and Figure 1: The sites selected for calibration, validation and testing mostly cluster in a specific area of climatic space. How representative are they of the whole European beech population? I know that data availability is a necessary driver of site selection, but something should be said about the limitations of the calibration. Having all countries represented does not seem relevant, as political boundaries are not climatic ones. It might have been better to include more northern sites or sites at the extremes.

Lines 175-177: Why was the relatively coarse-resolution dataset E-OBS used, rather than a newer and finer-resolution reanalysis product such as ERA5? This will affect the quality of the simulation

Lines 223-224: "Under a successful simulation we understand …" should be rephrased to "By 'successful simulation' we mean the ecosystem existence was maintained from spinup to the end of the normal run …".

Line 226: "maintain ecosystem existence" is an odd phrase—I assume that unsuccessful simulations involve complete mortality of plants only. I think it would be better to say "ecosystem stability". It's also not entirely clear what you mean by "potential causes"—are these input data, parameters, other model constraints?

Line 245: Why were AbgwC, SoilC and LitterC chosen as the output variables to examine? Why not fluxes, e.g. net primary production (NPP) or net ecosystem production (NEP)?

Lines 252-259: This paragraph is not clear. Provide more explanation of the global SA for readers not familiar with reference Verbeeck et al 2006 and "least square linearization method". Provide a link to RBBGCMuSo package in code and data availability section.

Line 282: Are estimation errors truly independent from each other? If they are not, what impact would this have on your parameter estimation?

Lines 279-292: This paragraph is a little hard to follow. Perhaps move the figures into the main text. Were there any cases where there was no overlap between parameters/sites? It is not necessarily a given that there exists a parameter set that leads to plausible simulations at all sites.

Section 2.5.1: It is not clear from the text what the DT method is or why it is being used here. It seems like a repeat of the steps described in previous sections. Is this an alternative method for selecting parameters? Or does it add to the previous analysis? A figure might be helpful to explain it.

Line 329: what "environmental conditions" did you test? I see it is in the results section but you should also mention specifics here.

3. Results

General: This section would benefit from a figure with a map showing the values of the 3 target output variables over the whole domain

3.1    Parameter sensitivity analysis: the text of this section could be expanded to explain in more detail why some parameters that seemingly have a high sensitivity in Figure 3 (e.g., GRC) were not chosen for calibration, whereas some that were chosen (e.g., Sseff) have a relatively lower sensitivity. Also, I find it surprisingly that all three output variables are highly sensitive to the same parameters. Is there a structural explanation for this?

Figure 4: The spread/distribution in site-specific optimized parameters sometimes cluster far away from the multi-site optimised value. This deserves more discussion.

Line 405: describe and/or provide a reference for the "Wilcoxon signed rank test"

Lines 435-436: You say that CLEC and MSC are significantly correlated. Does this have implications for your method or results?

Figure 7: most sites consistently increase or decrease, but CZ4 suddenly declines around 2015 and then recovers somewhat. Is there a reason for this?

4. Discussion

Line 495: Why would the use of parameters in previous model versions would prevent you from changing them in this one?

Lines 510-511: "Already" should not be used to start the sentence; rephrase

Line 522: insert "and" before "frequently"

Lines 537-540: The discussion of Bayesian calibration techniques feels too brief and out of place here; either discuss in more details the advantages and disadvantages of the technique compared to yours, or don't mention it

Line 543: what do you mean by "hybrid" approach?

Section 4.3: This analysis is good but it covers several different topics (parameter covariance with other parameters, parameter correlation with site characteristics) that should be separated into their own section. I would also suggest making the analysis more focused on the implications for model development beyond this specific application. You touch on this at the end, but do you results suggest that processes are missing and/or represented incorrectly in the model? What are priorities for future model development?

Lines 576: It's unclear how you simulated a varying CLEC. Please describe in more detail (perhaps in Methods section)

Lines 578-584: covariance/interdependency of parameters is an important point and should be discussed thoroughly

Line 607: Using a global database would seemingly contradict the need for a site-specific parameter! I think the lesson/recommendation is to measure these values routinely at research plots.

Section 4.4: This section is too long. I suggest breaking it further into subsections by output variable or process

Line 610: what do you mean by "large scale"? 100s, 1000s of kilometers? Different climatic zones rather than distances?

Line 613: are these statistics among simulated sites? This is where it would be helpful to have a spatial plot

Line 645: Are there other models that do include the process of soil aggregates? Has it been shown to improve model performance? Would this be a priority for BiomeBGCMuSo future development?

Lines 655-669: again what does this suggest about processes that are missing or inadequately represented in the model?

Line 670: should be "Similar" rather than "Similarly"

Line 686: delete "the" before "human influence"

Lines 739-748: This paragraph gets at what should be the key point of the paper, which is what can we learn about missing or inadequate representation of processes that are necessary for a more accurate simulation. This should be elaborated.

5. Conclusion

The conclusion is a little repetitive. I would like to see a discussion of priorities for future model development.

Code and data availability:

The input files used to run the model should be made available if possible, along with the optimization code

Supplement:

Inconsistent labeling of figures—in the text they are referred to as Fig A1, A2, etc but in the supplement they are labelled as Fig S1, S2, etc. This is also confusing with the labeling in Figure 2 to represent the steps in the optimization method (S1, S2, S3, S4). I recommend choosing a different notation in Figure 2 to avoid confusion

Figure S3: increase the size of the axes labels and ticks, it is not legible in its current form

Figure S6: I don't understand why there is more than one simulated and observed point for each parameter set. What is varying? Is it over time?

Figure S14: Missing reference in the caption

Figure S19: Missing reference in the caption. I suggest moving this to the main text as there is quite a bit of discussion around it.

---

## Referee Comment (RC2)

Review of „Biogeochemical model Biome-BGCMuSo v6.2 provides plausible and accurate simulations of carbon cycle in Central European beech forests"

In this paper the authors describe a method for calibrating a set of parameters for European beech. Different sets of plots were used for calibrating and validating, parameters were optimized for single sites or for multiple sites. The resulting parameters were compared with available values in literature, with a focus on carbon stock in aboveground wood biomass, soil and litter. The paper clearly presents a method to deal with the large amount of parameters many process-based models have.

**Some general comments:**
The discussion section is very long and very much focused on interdepencies and trends along stand and site gradients. Would it be an option to put all this material in a separate section (with clear subsections) and focus the discussion on the consequences for model development? Maybe then the discussion and conclusions can even be merged, since right now the conclusion contains (sometimes literal) repetitions of earlier mentioned topics. The very last paragraph of the discussion (starting line 739) is what I actually expected in this section.

I think it would be interesting to discuss a bit more in depth possible interdepencies between parameters and/or variables. Interdependencies between parameters is mentioned in line 578-579, but possible implications are not really discussed. Also, a „tight relationship between AbgwC and SoilC" is mentioned (line 517) and many parameters influence all three studied variables (Figure 3). It would be interesting to discuss the possible consequences of these interactions and the advantages/disadvantages of the proposed method in this context.

**Further comments**
Line 91: which elements?
Line 123: simulating more realistic dynamics (remove „a")
Line 124: again, using dynamic annual mortality rates (remove „a")
Line 132: simulations of crop functioning
Also, to line 132, is this relevant for the forest modelling? Are the crop parameters playing a role for forest modelling or are they completely separate?
Lines 176-180: I don't fully understand what you mean by extrapolating the E-OBS data. Do you mean that not all necessary input variables are measured and you use the gridded data to fill the gaps? Or are there sites without observations and that is what you use the E-OBS data for?
Line 244: sensitivity analysis (SA) (introduce abbreviation here)
Line 252: change brackets around Verbeeck et al 2006
Line 308-310: were the sites also selected to ensure a range of all the mentioned environmental conditions? I only saw a figure of the temperature and precipitation variability.
Line 501-502: the sentence feels a bit awkard, maybe it is meant like this: would cause a significant deviation from its mean or median of experimental observations.
Also, if the parameter should not be changed according to observations, does that indicate a process in the model itself should be improved?
Line 511: change brackets around Thronton et al 2002
Line 516: from A very tight positive relationship
Line 520-523: the sentence is a bit awkward, please rephrase
Line 541-542: awkard phrasing, please change (I think the term „averaging out effect" can be omitted)
Line 543: what is a hybrid approach?
Line 563: remove „the" → with decreasing plant density
line 565: change brackets around Timlin et al 2014

Lines 568-574: is this paragraph necessary?
Line 578: interdependencies
Line 605: change brackets around Thronton et al 2002
Line 731: Fix citation Kolb 2022
Line 754: any process-based model (singular)

---

## Author Comment (AC1)

*We would like to thank the reviewers for their valuable comments. Below please find our answers on the questions and comments raised by the Reviewer in detail. We also attach the revised manuscript (MS) highlighting the changes performed in the light of the comments received.*

*Note that the comments of Anonymous Referee #1 are shown below in bold with C (=Comment) letter at the beginning of each comment. Our responses to the comments are presented below in italic with R (=Reply) letter at the beginning of each reply.*

*We hope the revised version of the paper with the implemented corrections and modifications based on reviewers´ comments is now clear and understandable.*

**C: Review of Merganičová et al., "Biogeochemical model Biome-BGCMuSo v6.2 provides plausible and accurate simulations of carbon cycle in Central European beech forests"**

**C: General comments:**

**This paper describes a method of parameterization and application of the Biome-BGCMuSo v6.2 model for European beech forests and evaluates its accuracy in simulating aboveground wood, soil, and litter carbon stocks. In general, the paper is well-written and offers a useful methodology for setting parameters in biogeochemical models. It is very challenging to model carbon stocks correctly on a continental scale, and the authors have produced reasonable results for a large area in eastern Europe spanning multiple climatic zones. However, there are a few places where the manuscript could benefit from clarification and/or expanded discussion. I would like to see a section that clearly outlines the advantages and disadvantages of the parameterization methodology. It is currently discussed but scattered in various parts of the manuscript. It would also be good to focus the discussion more on general lessons—what processes are most important for accurate simulations, and what's missing or inadequately represented in the current version of the model— and priorities for future model development. There are also some minor corrections needed as specified below.**

*R:  Thank you for the feedback and valuable comments that considerably improved the manuscript. We have tried to clarify the unclear parts of the text, expanded the discussion based on the reviewers´ comments, and modified the conclusion to outline the advantages and disadvantages of the calibration method.*

**C: Specific comments:**

**C: Abstract**

**C: Line 28: "model wide-ranging applicability" sounds awkward; rephrase to something like, "applicability of the model over a wide range".**

*R:  Accepted, the wording has been changed accordingly.*

**C: 1**

**C: Introduction**

**C: The first few paragraphs (lines 52 – 71) contain a lot of generalizations and model development history that are not particularly helpful in introducing the topic of the manuscript. I suggest you cut most of this and focus specifically on the challenges of simulating carbon dynamics in forest ecosystems and different approaches for calibrating parameters.**

*R:  Accepted, the suggested text at the beginning  was deleted. We have slightly modified the calibration part . In our previous paper by Hollós et al. (2022) a detailed description of all methods with advantages and disadvantages is presented, so we do not want to repeat it again here. We hope in the revised text the emphasis is there with shorter but more focused introduction.*

**C:  Line 74: "intrinsic variability" isn't quite right here, you are describing uncertainty and range that comes from mostly external factors that are not intrinsic to the system**

*R:  Accepted, the word "intrinsic" was deleted and the word "own" was used instead.*

**C: Lines 77-79: awkward sentence, rephrase**

*R:  Accepted, the sentence was modified.*

**C: Lines 86-87: replace "generality" with "wide applicability"**

*R:  Accepted, the wording has been changed accordingly.*

**C:  Lines 86-87:  elaborate on the "advanced calibration designs"—what needs improvement, and what advantages does your approach offer?**

*R:  Sorry for the misleading wording. Our work offers improvements beyond the traditional methods that are detailed in Keenan et al., 2011 and Wallach et al., 2021. We adjusted the text to make it clear. ORIGINAL TEXT: Therefore, advanced calibration designs maintaining a balance between the local accuracy and generality, and providing good performance across multiple variables, have been developed (Keenan et al., 2011; Wallach et al., 2021)."*

*REVISED TEXT:*

*"Therefore, introducing advanced calibration designs that are supposed to maintaining a balance between the local accuracy and generalitywide applicability, and providing good performance across multiple variables, is needed, have been developed (Keenan et al., 2011; Wallach et al., 2021). In this study, we focused on creating a calibration workflow that offers improvements beyond the traditional methods detailed in (Keenan et al., (2011), and; Wallach et al., (2021)."*

**C:  Line 89: replace "addressed" with "used"**

*R:  Accepted, the wording has been changed accordingly.*

**C:  Line 91: "these elements"—specific what these are (e.g. carbon, nitrogen, etc)**

*R:  Accepted, the text was modified accordingly.*

**C:  Line 97: delete "the" at end of line**

*R:  Accepted, "the" has been deleted.*

**C:  Line 105: not sure what is meant by "phenotypical plasticity", please explain/elaborate**

*R:  We added an explanation in the text.*

**C:  2**

**C:  Data and Methods**

**C:  2.1 Model: This section would benefit from the addition of more details about the model, such as the timestep (daily) and the main inputs (meteorology and soil information). While the next subsection 2.2 mentions the datasets used, it's not clear from the text that these are required to drive the model.**

*R:  Accepted, we added more details about the model in the text.*

**C:  Lines 116-117: It's not clear what is meant by a "two-leaf" representation**

*R:  We added the explanation in the text: "... i.e. assuming one sunlit and one shaded leaf to represent stand foliage"*

**C:  Line 128: "storages" should be singular, i.e. "storage"**

*R:  Accepted, the wording has been changed accordingly.*

**C:  Lines 130-132: Here you are implying that there are many crop-specific parameters that were added, but are these needed to simulate forests?**

*R:  No, they are not needed for forests, we modified the text to clarify this.*

**C:  Lines 150- 157 and Figure 1: The sites selected for calibration, validation and testing mostly cluster in a specific area of climatic space. How representative are they of the whole European beech population? I know that data availability is a necessary driver of site selection, but something should be said about the limitations of the calibration. Having all countries represented does not seem relevant, as political boundaries are not climatic ones. It might have been better to include more northern sites or sites at the extremes.**

*R:  We agree with the reviewer that the data availability drives the selection of sites. We included more information about data and representation of beech population in Data and Discussion, where we also discussed the limitations of site selection.*

**C:  Lines 175-177: Why was the relatively coarse-resolution dataset E-OBS used, rather than a newer and finer-resolution reanalysis product such as ERA5? This will affect the quality of the simulation**

*R:  There are important differences between the E-OBS and the ERA5 datasets that were considered during the selection of the driving meteorological database. Focusing on the spatial resolution, in fact E-OBS is comparable with ERA5-Land (both have 0.1x0.1 degree resolution), but the original ERA5 (that is mentioned by the Reviewer) has a coarser resolution (0.25x0.25 degree). Considering the information content of the databases the differences are more relevant. ERA5 and ERA5-Land are reanalyses, which*

means that they are not directly based on observations, but rather the disseminated data is constructed with meteorological data assimilation and short range forecasts using a numerical weather prediction (NWP) model. It means that station data is used for temperature, but the data assimilation also uses prior information from previous short range forecasts, so that the resulting reanalysis data is a special combination of observations and models. For precipitation (that is an important input data of the Biome-BGCMuSo model) the situation is different. ERA5/ERA5-Land-based precipitation is calculated entirely by the NWP, as assimilation of observed precipitation data is impossible in the NWP models. This means that precipitation observations are not used in ERA5/ERA5-Land. Reanalysis based precipitation is comparable to observations since the quality of the short range precipitation forecasts for Europe is quite good, but still, the disseminated precipitation data is entirely modelled and not observed. In contrast, E-OBS was constructed from observed temperature and precipitation via interpolation, which means that the dataset is directly derived from the observations without any model included (interpolation can be considered as a model, but that is inevitable in case of gridded meteorological datasets). In summary, E-OBS was preferred due to its good spatial resolution and the method how the dataset was constructed.

C:  Lines 223-224: "Under a successful simulation we understand …" should be rephrased to "By 'successful simulation' we mean the ecosystem existence was maintained from spinup to the end of the normal run …".

R:  Accepted, the wording has been changed accordingly.

C:  Line 226: "maintain ecosystem existence" is an odd phrase—I assume that unsuccessful simulations involve complete mortality of plants only. I think it would be better to say "ecosystem stability". It's also not entirely clear what you mean by "potential causes"—are these input data, parameters, other model constraints?

R:  Thank you for the comment. Yes, it is true that the unsuccessful simulations mean complete mortality of plants only. We accepted your suggestion and used the term "ecosystem stability" instead. We also explained what we meant under "potential causes" - from the point of model calibration they are parameter values.

C:  Line 245: Why were AbgwC, SoilC and LitterC chosen as the output variables to examine? Why not fluxes, e.g. net primary production (NPP) or net ecosystem production (NEP)?

R:  We explained the reasons in the text. The reasons behind are the data availability to cover larger gradients of environmental conditions than fluxes which are usually available only for few sites. Moreover, the long-term time series of AbgwC covering up to 69-year-long periods also allow the evaluation of the simulated temporal development over time.

C:  Lines 252-259: This paragraph is not clear. Provide more explanation of the global SA for readers not familiar with reference Verbeeck et al 2006 and "least square linearization method". Provide a link to RBBGCMuSo package in code and data availability section.

R:  Thank you, we modified the text to clarify the approach and added the link to RBBGCMuSo package in the code section.

C:  Line 282: Are estimation errors truly independent from each other? If they are not, what impact would this have on your parameter estimation?

R:  The analyses of the simulations showed that the estimation errors are not independent in the tested parameter space, see Fig. S8. This is because all three examined variables are a part of the carbon cycle. Due to their interdependencies, the site-specific calibration at some sites did not lead to reduced errors of all tested variables considering the selected parameters as we pointed it out in Discussion. Nevertheless, the interdependencies may be accounted for by including a covariance matrix into a multivariate likelihood function, which will however further increase the computational complexity of the method, already characterized by its significant demands.

C:  Lines 279-292: This paragraph is a little hard to follow. Perhaps move the figures into the main text. Were there any cases where there was no overlap between parameters/sites? It is not necessarily a given that there exists a parameter set that leads to plausible simulations at all sites.

R:  Thanks, we moved the figures to the main text and reformulated some parts of the text to clarify the issue. Yes, it is true, that no overlap could occur, however in our study we did not encounter such cases

*under the specified contraints of output variables. However, if such a case happened, we would apply the rule of majority, as we did in the case of decision trees.*

**C: Section 2.5.1: It is not clear from the text what the DT method is or why it is being used here. It seems like a repeat of the steps described in previous sections. Is this an alternative method for selecting parameters? Or does it add to the previous analysis? A figure might be helpful to explain it.**

*R: Our aim with DT was to examine the robustness of the optimised SSMV and MSMV parameters. This was done using the machine learning method of DT that identifies ranges of parameters within which there is a high probability that simulations will be within the plausible ranges of output variables. Yes, the approach of processing the data is similar, but the definition of plausible ranges is based on a different mathematical algorithm. We added a figure to clarify the method.*

**C: Line 329: what "environmental conditions" did you test? I see it is in the results section but you should also mention specifics here.**

*R: Accepted, we added this information in the text.*

**C: 3**

**C: Results**

**C: General: This section would benefit from a figure with a map showing the values of the 3 target output variables over the whole domain**

*R: Thanks, we added a figure with suggested maps (new Figure 10).*

**C: 3.1**

**C: Parameter sensitivity analysis: the text of this section could be expanded to explain in more detail why some parameters that seemingly have a high sensitivity in Figure 3 (e.g., GRC) were not chosen for calibration, whereas some that were chosen (e.g., Sseff) have a relatively lower sensitivity. Also, I find it surprisingly that all three output variables are highly sensitive to the same parameters. Is there a structural explanation for this?**

*R: Thank you for the comment. This information can be found in the discussion. The three output variables are sensitive to the same parameters because they are all a part of the carbon cycle.*

**C: Figure 4: The spread/distribution in site-specific optimized parameters sometimes cluster far away from the multi-site optimised value. This deserves more discussion.**

*R: We added a paragraph in data and discussion section about this issue.*

**C: Line 405: describe and/or provide a reference for the "Wilcoxon signed rank test"**

*R: Accepted, we added the information in the text.*

**C: Lines 435-436: You say that CLEC and MSC are significantly correlated. Does this have implications for your method or results?**

*R: The covariance of parameters was revealed after the calibration and implies this relationship, if proven biologically, can be used in future calibration efforts, when only one of the correlated parameters would be selected while the other one can be derived. The Monte Carlo procedure used during the calibration ensures that the selection of parameter values was not affected by this relationship.*

**C: Figure 7: most sites consistently increase or decrease, but CZ4 suddenly declines around 2015 and then recovers somewhat. Is there a reason for this?**

*R: There was a strong management intervention at the research site that was also considered in simulations.*

**C: 4**

**C: Discussion**

**C: Line 495: Why would the use of parameters in previous model versions would prevent you from changing them in this one?**

*R: Thank you for the comment, we have chosen this approach because of many new parameters included in the calibrated model version and high demand on calculation time and storage space. However, that does not hinder their future modification if supported by empirical evidence.*

**C: Lines 510-511: "Already" should not be used to start the sentence; rephrase**

*R: Accepted, the wording has been changed accordingly.*

**C: Line 522: insert "and" before "frequently"**

*R: Accepted, the wording has been changed accordingly.*

**C: Lines 537-540: The discussion of Bayesian calibration techniques feels too brief and out of place here; either discuss in more details the advantages and disadvantages of the technique compared to yours, or don't mention it**

*R: We modified the text. The modified text reads as follows: This problem can be mitigated using calibration techniques which utilize a more appropriate likelihood function (i.e. formal likelihood; Hollós et al., 2022). In this way, the issue of the significant spatial autocorrelation can be resolved using the inclusion of an appropriate error-covariance matrix in the construction of the likelihood function (Tarantola, 2005).*

**C: Line 543: what do you mean by "hybrid" approach?**

*R: We explained the approach in more detail.*

**C: Section 4.3: This analysis is good but it covers several different topics (parameter covariance with other parameters, parameter correlation with site characteristics) that should be separated into their own section. I would also suggest making the analysis more focused on the implications for model development beyond this specific application. You touch on this at the end, but do you results suggest that processes are missing and/or represented incorrectly in the model? What are priorities for future model development?**

*R: We divided the section 4.3: into two subsections 4.3.1. Covariance of parameters, and 4.3.2 Parameter correlations with site characteristics, and added a new section 4.5. Future model development, where we discussed further work.*

**C: Lines 576: It's unclear how you simulated a varying CLEC. Please describe in more detail (perhaps in Methods section)**

*R: Accepted, we added an explanation in Methods section, subsetion 2.5.1.*

**C: Lines 578-584: covariance/interdependency of parameters is an important point and should be discussed thoroughly**

*R: Accepted, we created a separate section and included additional text dealing with the issue.*

**C: Line 607: Using a global database would seemingly contradict the need for a site-specific parameter! I think the lesson/recommendation is to measure these values routinely at research plots.**

*R: Thank you for the comment. Of course, locally measured values are most valuable, however if those are not available, then global databases may be helpful, as various relationships can be derived from their data that could consider local environmental conditions. I tried to rephrase the text to clarify this.*

**C: Section 4.4: This section is too long. I suggest breaking it further into subsections by output variable or process**

*R: Accepted, the section was divided into several subsections based on the main output variable discussed.*

**C: Line 610: what do you mean by "large scale"? 100s, 1000s of kilometers? Different climatic zones rather than distances?**

*R: Under the large scale we understand here the level of the whole geographical region of interest. We clarified it in Sect. 2.5.3.*

**C: Line 613: are these statistics among simulated sites? This is where it would be helpful to have a spatial plot**

*R: yes, they are. We added the maps in the Results section and hope the results are now more understandable. Nevertheless, please note that the maps represent only one time point in simulations whereas the statistics were calculated from the whole time series.*

**C: Line 645: Are there other models that do include the process of soil aggregates? Has it been shown to improve model performance? Would this be a priority for BiomeBGCMuSo future development?**

*R: We are not aware of any model that includes this process. At the moment it is not a priority for future model development, because more tests are needed including different ecosystems and sites that would cover the whole range of clay proportion .*

**C: Lines 655-669: again what does this suggest about processes that are missing or inadequately represented in the model?**

*R:  The findings indicate decomposition may not be adequatelly captured in the model yet. However,a more detailed analysis of this process is needed to confirm these findings. We added  text summarising our results and possible reasons.*

**C:  Line 670: should be "Similar" rather than "Similarly"**

*R: Based on the consultation with a native speaker, we did not change this word.*

**C:  Line 686: delete "the" before "human influence"**

*R:  Accepted, "the" has been deleted.*

**C:  Lines 739-748: This paragraph gets at what should be the key point of the paper, which is what can we learn about missing or inadequate representation of processes that are necessary for a more accurate simulation. This should be elaborated.**

*R:  We created a new chapter 4.5. devoted to future model development and elaborated the topic in more detail.*

**C:  5**

**C:  Conclusion**

**C:  The conclusion is a little repetitive. I would like to see a discussion of priorities for future model development.**

*R:  We modified the conclusion aand added parts discussion advantages and disadvantages of the calibration method.*

**C:  Code and data availability:**

**C:  The input files used to run the model should be made available if possible, along with the optimization code**

*R:  The initial data and the code for optimisation were addedin the supplement.*

**C:  Supplement:**

**C:  Inconsistent labeling of figures—in the text they are referred to as Fig A1, A2, etc but in the supplement they are labelled as Fig S1, S2, etc. This is also confusing with the labeling in Figure 2 to represent the steps in the optimization method (S1, S2, S3, S4). I recommend choosing a different notation in Figure 2 to avoid confusion**

*R:  Thank you for the comment. We changed the labelling accordingly. For phases in Figure2 we applied WS, and for Supplement S.*

**C:  Figure S3: increase the size of the axes labels and ticks, it is not legible in its current form**

*R:  We increased the size of labels and hope it is readable now.*

**C:  Figure S6: I don't understand why there is more than one simulated and observed point for each parameter set. What is varying? Is it over time?**

*R:  Thank you for the comment. Yes, the graph shows the time series. We changed the description to clarify the issue*

**C:  Figure S14: Missing reference in the caption**

*R:  Accepted, the reference was added.*

**C:  Figure S19: Missing reference in the caption. I suggest moving this to the main text as there is quite a bit of discussion around it.**

*R:  Thank you for the comment. We corrected the caption. We carefully considered the suggestion to move the figure to the main text, but decided to leave it in supplement due to the sizes of both the manuscript and the figure. The supplement is easily accessible to all interested readers.*

---

## Author Comment (AC2)

*We would like to thank the reviewers for their valuable comments. Below please find our answers on the questions and comments raised by the Reviewer in detail. We also attach the revised manuscript (MS) highlighting the changes performed in the light of the comments received.*

*Note that the comments of Anonymous Referee #1 are shown below in bold with C (=Comment) letter at the beginning of each comment. Our responses to the comments are presented below in italic with R (=Reply) letter at the beginning of each reply.*

*We hope the revised version of the paper with the implemented corrections and modifications based on reviewers´ comments is now clear and understandable.*

**C: Review of „Biogeochemical model Biome-BGCMuSo v6.2 provides plausible and accurate simulations of carbon cycle in Central European beech forests"**

**C: In this paper the authors describe a method for calibrating a set of parameters for European beech. Different sets of plots were used for calibrating and validating, parameters were optimized for single sites or for multiple sites. The resulting parameters were compared with available values in literature, with a focus on carbon stock in aboveground wood biomass, soil and litter. The paper clearly presents a method to deal with the large amount of parameters many process-based models have.**

*R: Thank you for the thorough review of our paper and your valuable comments that considerably improved the manuscript. We have tried to clarify the unclear parts of the text, expanded the discussion based on the reviewers´ comments, and modified the conclusion to outline the advantages and disadvantages of the calibration method.*

**C: Some general comments:**
**C: The discussion section is very long and very much focused on interdepencies and trends along stand and site gradients. Would it be an option to put all this material in a separate section (with clear subsections) and focus the discussion on the consequences for model development?**
*R: Accepted, the discussion was divided into several subsections as suggested by both reviewers.*
**C: Maybe then the discussion and conclusions can even be merged, since right now the conclusion contains (sometimes literal) repetitions of earlier mentioned topics. The very last paragraph of the discussion (starting line 739) is what I actually expected in this section.**
*R: We created a new chapter 4.5. devoted to future model development and elaborated the topic in more detail. We also modified the conclusion where we focused on advantages and disadvantages of the calibration method.*

**C: I think it would be interesting to discuss a bit more in depth possible interdepencies between parameters and/or variables. Interdependencies between parameters is mentioned in line 578-579, but possible implications are not really discussed.**
*R: We created an additional subsection 4.3.1 dealing with the interdependencies between parameters and included additional text. We discussed possible implications at the end of subsection 4.3.1. as follows: "Our results indicate the necessity of analysing the covariance between parameters during a model calibration as it not only enlightens the model behaviour and interdependencies between specific parameters but can also increase the efficiency of the calibration procedure by excluding one of the correlated parameters from the calibrated parameter set and estimating its value only subsequently. In addition, such information may also help to identify the gaps in the available empirical evidence and the direction of future empirical research. "*
**C: Also, a „tight relationship between AbgwC and SoilC" is mentioned (line 517) and many parameters influence all three studied variables (Figure 3). It would be interesting to discuss the possible consequences of these interactions and the advantages/disadvantages of the proposed method in this context.**
*R: We added a paragraph at the beginning of Section 4.1 and in Section 4.2, where we discuss this issue in more detail.*

**C: Further comments**

**C: Line 91: which elements?**

*R: We added the information in the text.*

**C: Line 123: simulating more realistic dynamics (remove „a")**

*R: Accepted, "a" has been deleted.*

**C: Line 124: again, using dynamic annual mortality rates (remove „a")**

*R: Accepted, "a" has been deleted.*

**C: Line 132: simulations of crop functioning**

*R: Accepted, "the" has been deleted.*

**C: Also, to line 132, is this relevant for the forest modelling? Are the crop parameters playing a role for forest modelling or are they completely separate?**

*R: No, crop functioning parameters are not needed for forests, we modified the text to clarify this.*

**C: Lines 176-180: I don't fully understand what you mean by extrapolating the E-OBS data. Do you mean that not all necessary input variables are measured and you use the gridded data to fill the gaps? Or are there sites without observations and that is what you use the E-OBS data for?**

*R: We reformulated the sentence to clarify the issue. Yes, both of the mentioned cases occurred in our data. In some cases we had to fill the gaps in the measured data, while in other cases no measurements were available, and hence we used E-OBS data. In addition, in E-OBS data, some climate variables required by our model are missing. These were calculated by MTCLIM.*

**C: Line 244: sensitivity analysis (SA) (introduce abbreviation here)**

*R: Accepted, the abbreviation was introduced.*

**C: Line 252: change brackets around Verbeeck et al 2006**

*R: Accepted, the text was changed accordingly.*

**C: Line 308-310: were the sites also selected to ensure a range of all the mentioned environmental conditions? I only saw a figure of the temperature and precipitation variability.**

*R: Yes, all available information about environmental codnitions were considered, including elevation, soil characteristics, as shown in Table 1. We added the information in the text.*

**C: Line 501-502: the sentence feels a bit awkard, maybe it is meant like this: would cause a significant deviation from its mean or median of experimental observations.**

*R: Accepted, the text was changed accordingly.*

**C: Also, if the parameter should not be changed according to observations, does that indicate a process in the model itself should be improved?**

*R: We tried to clarify the issue. If observations are available, parameters should definitely be changed based on the empirical evidence, particularly if we are focused at a local scale. Experiments at a larger scale should also consider observations, but if we want to use a generic parameter set, then the optimised value for the whole region may not be equal to observations from a single site.*

*From our results it does not seem that any particular process is completely wrong, although we identified some processes that may benefit from improvements in the model. We summarised them in Section 4.5.*

**C: Line 511: change brackets around Thronton et al 2002**

*R: Accepted, the text was changed accordingly.*

**C: Line 516: from A very tight positive relationship**

*R: Accepted, "a" has been added.*

**C: Line 520-523: the sentence is a bit awkward, please rephrase**

*R: Accepted, the text was changed.*

**C: Line 541-542: awkard phrasing, please change (I think the term „averaging out effect" can be omitted)**

*R: Accepted, the text was changed accordingly.*

**C: Line 543: what is a hybrid approach?**

*R: We explained the approach in more detail.*

**C: Line 563: remove „the" → with decreasing plant density**

*R: Accepted, "the" has been deleted.*

**C:** **line 565: change brackets around Timlin et al 2014**

*R:* *Accepted, the text was changed accordingly.*

**C:** **Lines 568-574: is this paragraph necessary?**

*R:* *We believe the information may be useful for other modellers using different models.*

**C:** **Line 578: interdependencies**

*R:* *Accepted, the spelling has been corrected.*

**C:** **Line 605: change brackets around Thronton et al 2002**

*R:* *Accepted, the text was changed accordingly.*

**C:** **Line 731: Fix citation Kolb 2022**

*R:* *Accepted, the text was changed accordingly.*

**C:** **Line 754: any process-based model (singular)**

*R:* *Accepted, the singular is used.*
